# Programmable mutually exclusive alternative splicing for generating RNA and protein diversity

Melina Mathur[1], Cameron M. Kim[1], Sarah A. Munro[1,2,3,5], Shireen S. Rudina[1], Eric M. Sawyer [1,6] & Christina D. Smolke[1,4]

Alternative splicing performs a central role in expanding genomic coding capacity and proteomic diversity. However, programming of splicing patterns in engineered biological systems remains underused. Synthetic approaches thus far have predominantly focused on controlling expression of a single protein through alternative splicing. Here, we describe a modular and extensible platform for regulating four programmable exons that undergo a mutually exclusive alternative splicing event to generate multiple functionally-distinct proteins. We present an intron framework that enforces the mutual exclusivity of two internal exons and demonstrate a graded series of consensus sequence elements of varying strengths that set the ratio of two mutually exclusive isoforms. We apply this framework to program the DNA-binding domains of modular transcription factors to differentially control downstream gene activation. This splicing platform advances an approach for generating diverse isoforms and can ultimately be applied to program modular proteins and increase coding capacity of synthetic biological systems.

[1] Department of Bioengineering, Stanford University, Stanford, CA 94305, USA. [2] Joint Initiative for Metrology in Biology, Stanford, CA 94305, USA. [3] Genome-scale Measurements Group, National Institute of Standards and Technology, Stanford, CA 94305, USA. [4] Chan Zuckerberg Biohub, San Francisco, CA 94158, USA. [5] Present address: Minnesota Supercomputing Institute, University of Minnesota, Minneapolis, MN 55455, USA. [6] Present address: Department of Molecular and Cell Biology, University of California, Berkeley, Berkeley, CA 94720, USA. Correspondence and requests for materials should be addressed to C.D.S. (email: csmolke@stanford.edu)

Recent developments in synthetic biology have enabled numerous techniques for programming cellular functions in industrial biotechnology[1], agriculture[2], and medicine[3–5]. Powerful genetic tools, such as CRISPR/Cas9 gene editing[6], microRNA gene silencing[7], and epigenetic remodeling[8–10], have been pivotal in expanding synthetic biology from prokaryotic and lower eukaryotic cells into mammalian systems. Despite these advances, alternative splicing, a critical eukaryotic regulatory mechanism, has been overlooked in synthetic biology. Alternative splicing serves an essential role in increasing biological complexity, genomic coding capacity, and proteomic diversity across developmental stages and cell types[11]. Over 95% of multi-exon genes in humans are alternatively spliced[12] and multi-exon human genes produce, on average, three or more distinct RNA isoforms[13]. Thus, alternative splicing holds immense potential to advance the scale and complexity of engineered biological systems.

Genetic devices incorporating alternative splicing mechanisms have been functionally limited to controlling expression of a single protein through coupling a gene to a splicing regulatory component. In one example, a three-exon, two-intron system uses exon skipping, in which inclusion of an alternative exon encoding a premature stop codon halts translation, resulting in a single expressed protein from one of two possible spliced isoforms[14]. In another example, a four-exon, three-intron system uses mutually exclusive alternative splicing (MEAS) to incorporate one of two internal exons into the final transcript[15]. MEAS devices of this type contain frameshift nucleotides in one of the mutually exclusive exons, such that splicing produces only one functional protein.

By extending regulated expression beyond one protein, we can advance alternative splicing-based approaches towards producing diverse protein isoforms from a single splicing event encoded by a genetic device. MEAS is particularly well suited for generating protein diversity by swapping exons without disrupting protein size or structure[16,17]. Natural biological systems use MEAS to control enzymes and complex proteins, such as by altering a regulatory region of pyruvate kinase (PKM) to modulate the final step of glycolysis[18] or by altering the pore-forming subunit of a calcium-activated potassium channel (KCNMA1) to modulate its gating characteristics[19]. Analogous alternative splicing capabilities in engineered biological systems have yet to be achieved.

We developed an alternative splicing platform for regulating exons that undergo MEAS to generate at least two RNA isoforms that translate into distinct functional proteins. We designed an intron framework with three introns and four exons for constructing alternative splicing devices (ASDs) that encode synthetic, modular, and programmable exon sequences within identified design constraints. We mutated intronic consensus sequence elements to develop a graded series of branchpoint (BP) and polypyrimidine tract (PPT) elements of different strengths that tune the relative abundance of two mutually exclusive RNA isoforms and resulting proteins. Finally, we encoded DNA-binding domains within programmable exons of the intron framework to produce transcription activator-like effector transcription factors (TALE-TFs) with varying DNA-binding functionalities in a mutually exclusive manner to differentially activate reporter genes. Our work demonstrates a programmable and extensible MEAS platform for generating functionally diverse RNA and protein isoforms.

## Results

### A mutually exclusive alternative splicing intron framework.
We developed a MEAS intron framework comprising three introns interspersed between four exon positions, of which the second and third exons (exons 2 and 3) are mutually exclusive (Fig. 1a). We harvested key genetic components capable of encoding a MEAS event from the minigene of the α-tropomyosin gene (Tpm1) in Rattus norvegicus[20]. Prior studies of this minigene revealed that the mutually exclusive behavior arises from a cis-directed mechanism encoded within the second intron[21]. A proximal 5′ splice site (ss) located 42 nucleotides upstream of the branchpoint (BP) sequence of exon 3 prevents spliceosome assembly across the intron between exons 2 and 3 such that these exons are not efficiently spliced together. Exon 3 is incorporated into the dominant spliced isoform by default in most cell types[22].

In our MEAS intron framework, we base the 5′ ss, 3′ ss, BP, and polypyrimidine tract (PPT) sequences required for exon–intron definition, spliceosome recruitment, and intron excision[23] on the corresponding sequences from the minigene (Fig. 1b). The intron framework has only one open reading frame, with a start codon at the beginning of exon 1 and a stop codon at the end of exon 4. We then established design criteria for programming exonic sequences. First, we set the length of the mutually exclusive exons to be ~126 nucleotides as in the minigene[20]. Second, to avoid splicing events that result in frameshifts, we designed exons with lengths in multiples of three nucleotides. Third, we defined exonic boundaries by setting splice site sequences to match those selected from the minigene when possible.

**Programmed exons in the framework splice to dominant isoform.** To validate the base intron framework and verify flexibility of exonic sequences, we constructed alternative splicing devices (ASDs) that regulate the mutually exclusive production of two fluorescent proteins, mCherry[24] and Clover[25], which have distinct excitation and emission spectra (Fig. 2a). The first device (ASD mCherry) encodes mCherry within exons 1 and 3, and Clover within exons 2 and 4 (Fig. 2b). The expected result of splicing to isoform 1-3-4 is mCherry fluorescence while the expected result of splicing to isoform 1-2-4 is Clover fluorescence. The first mCherry exon (exon 1) conserves the 5′ ss (CAG). The second mCherry exon (exon 3) maintains the 3′ ss (CTG) and the 5′ ss (GAT). While the first Clover exon (exon 2) conserves the 3′ ss (CTC) and the 5′ ss (AAG), maintaining the 3′ ss of the second Clover exon (exon 4) requires mutating a leucine codon (CTG) into an alanine codon (GCT). mCherry and Clover fluorescence are retained in spliced controls harboring this mutation (Supplementary Fig. 1).

The second device (ASD Clover) encodes Clover within exons 1 and 3, and mCherry within exons 2 and 4 (Fig. 2c). The expected result of splicing to isoform 1-3-4 is Clover fluorescence while the expected result of splicing to isoform 1-2-4 is mCherry fluorescence. Upon mutating the 5′ ss of exon 1 from a tyrosine codon (TAC) to a glutamine codon (CAG) and the 5′ ss of exon 2 from a glutamic acid codon (GAG) to a lysine codon (AAG), all splice sites are conserved. Fluorescence is retained in spliced controls with these mutations (Supplementary Fig. 2). We additionally attempted to modify the first three nucleotides in exon 4 from a glycine codon (GGC) to an alanine codon (GCT) to conserve the 3′ ss. However, this mutation abolished mCherry fluorescence and thus was not incorporated into ASD Clover (Supplementary Fig. 2).

Since naturally occurring exons generally contain abundant cis-acting exonic splicing enhancer (ESE) regulatory elements that promote exon recognition[26], we conducted a bioinformatic analysis of ESE and exonic splicing silencer (ESS) elements in the mutually exclusive exons of ASD mCherry and ASD Clover (Supplementary Fig. 3). In these fluorescent protein exons, ESE elements are comparatively stronger than ESS elements. We also

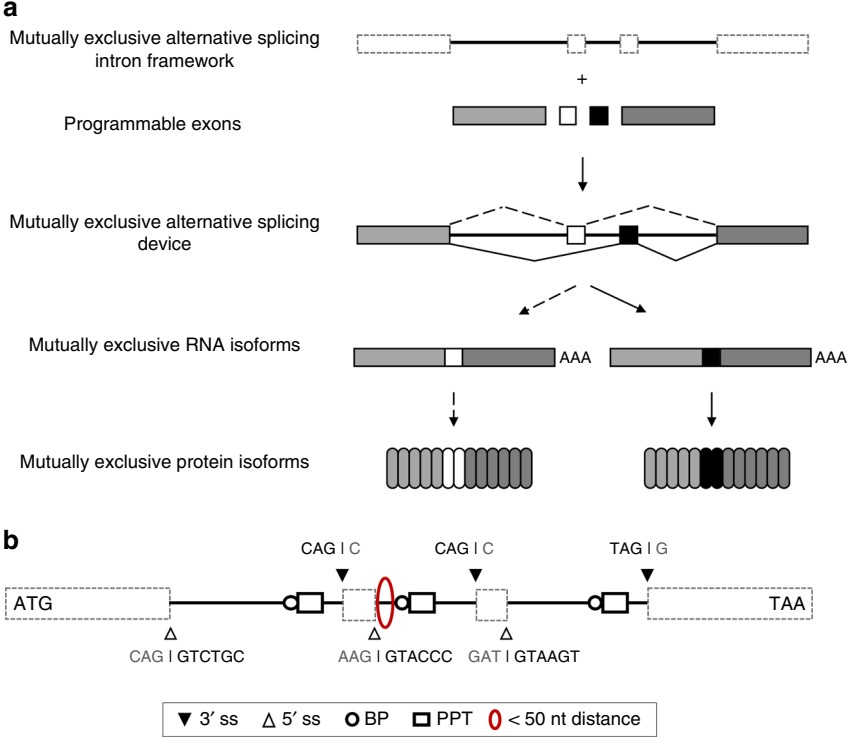

**Fig. 1** Overview of the MEAS intron framework. **a** The intron framework contains three introns that support MEAS events. Incorporation of four programmable and modular exons into the intron framework forms a MEAS device. The device produces at least two mutually exclusive RNA isoforms as a result of splicing, and these RNA sequences are translated into distinct protein isoforms. **b** Specific features of the intron framework enabling MEAS include: (i) 3′ ss (black triangles with native sequences noted above the triangles) and 5′ ss (white triangles with native sequences noted below the triangles) sequences define exon–intron junctions, (ii) BP sequences (white circles) and PPT sequences (white rectangles) facilitate splicing, and (iii) a proximal exon 2 5′ ss and exon 3 BP sequence separated by less than 50 nucleotides (red ellipse) prevents spliceosome assembly in order to enforce mutually exclusive behavior. The intron framework encodes one open reading frame with the illustrated start and stop codons

confirmed that a mutually exclusive exon length range of 111 to 150 nucleotides is tolerated in our design (Supplementary Fig. 4).

We examined ASD splicing patterns based on RNA and protein analyses. Plasmids encoding devices were transfected into HEK-293T cells and full-length RNA was extracted from cultured cells. We conducted long-read sequencing using a modified Pacific Biosciences Iso-Seq method to comprehensively characterize the RNA isoform profiles generated by the ASDs. This RNA analysis technique detects isoforms produced by each ASD in a hypothesis-free manner[27], enabling the observation of aberrant isoforms. Some minor aberrant isoforms may not be detectable in the sequenced RNA pools if they result in a premature termination codon in the transcript that activates nonsense-mediated decay[28].

RNA analysis of ASD mCherry indicates that the device generates two isoforms (Fig. 2d). Isoform 1-3-4 accounts for 98.3% of spliced RNA from ASD mCherry, while the remaining 1.7% represents an aberrant isoform that includes the first and second introns. Meanwhile, analysis of ASD Clover reveals production of three isoforms (Fig. 2e). The dominant isoform 1-3-4 accounts for 79.3% of spliced RNA. The second most prevalent spliced isoform accounts for 15.0% of spliced RNA and is an aberrant version of isoform 1-3-4 using a cryptic splice site in exon 4 located 171 nucleotides downstream of the defined 3′ ss. The remaining 5.7% of spliced RNA from ASD Clover represents an aberrant isoform that includes the third intron. RNA analysis of both ASDs indicates that isoform 1-3-4 is the dominant spliced product.

We then assessed the fluorescence of the spliced products from ASDs via flow cytometry. Four controls were built for each device to benchmark fluorescence levels of the spliced products: the mutually exclusive selection of exon 2 (isoform 1-2-4), the mutually exclusive selection of exon 3 (isoform 1-3-4), the double inclusion product (isoform 1-2-3-4), and the double exclusion product (isoform 1-4). The controls for ASD mCherry exhibit the expected fluorescence patterns: isoforms 1-2-3-4 and 1–4 exhibit low fluorescence, isoform 1-2-4 exhibits high Clover fluorescence, and isoform 1-3-4 exhibits high mCherry fluorescence (Fig. 2f). ASD mCherry results in mCherry fluorescence equivalent to that of the isoform 1-3-4 control. Although the spliced isoform from ASD mCherry encodes a fragment of the protein coding sequence of Clover, this fragment does not result in Clover fluorescence as validated by microscopy (Fig. 2h). Similar spliced controls for ASD Clover exhibit the expected fluorescence patterns: isoforms 1-2-3-4 and 1–4 exhibit low fluorescence, isoform 1-2-4 exhibits high mCherry fluorescence, and isoform 1-3-4 exhibits high Clover fluorescence. Clover fluorescence from ASD Clover is 25% lower than that of the isoform 1-3-4 control (Fig. 2g, i), possibly resulting from the two aberrant products revealed by long-read sequencing being transcribed less efficiently than the spliced control or from the fragment of the mCherry coding sequence in exon 4 producing a peptide interfering with folding and maturation of Clover. These results suggest that the intron framework supports programming of exonic sequences and that the base design generates isoform 1-3-4 as the most abundant isoform.

**Tunable consensus sequence elements alter isoform profiles.** Studies of the α-tropomyosin minigene show that the consensus BP and PPT sequences of exon 3 more strongly support

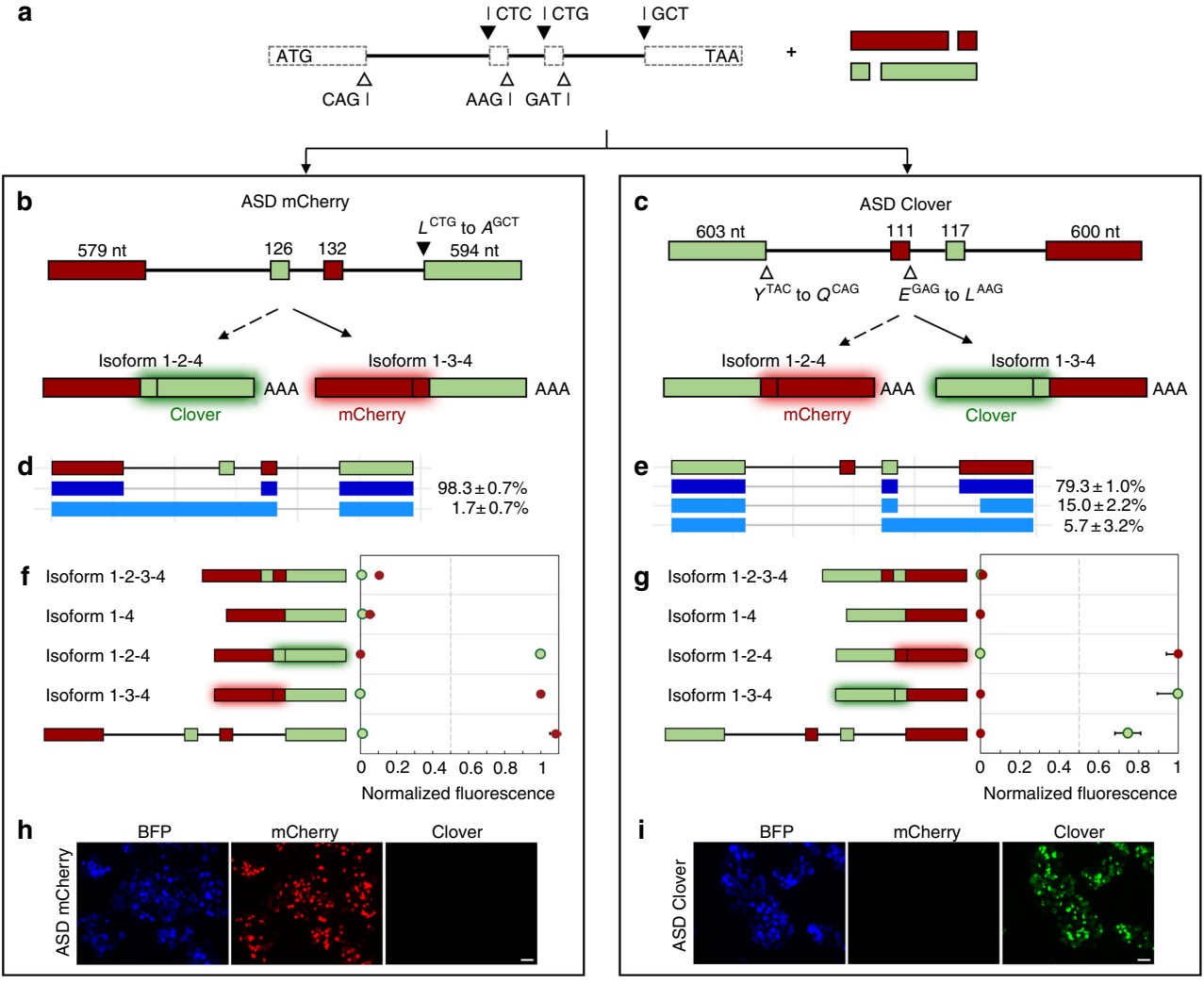

**Fig. 2** Design and validation of fluorescent MEAS devices. **a** Fluorescent ASDs are constructed using the base intron framework and synthetic fluorescent protein exons. The defined splice site sequences are incorporated into the programmable exons. The first codon of each exon (noted to the right of the black triangle) and last codon of each exon (noted to the left of the white triangle) is set to assure proper exon definition when possible. **b** ASD mCherry and **c** ASD Clover were designed to generate distinct fluorescent outputs as a result of a single MEAS event. The lengths, in nucleotides, of the exons are noted. Specific codons were set to maintain splice site sequences and their corresponding 1-letter amino acid codes are shown in italicized bold. Plasmids encoding each ASD or spliced isoform control with a BFP transfection marker were transfected into HEK-293T cells. RNA was extracted for characterization using long-read sequencing and protein levels were assayed for fluorescence using flow cytometry. RNA isoform profiles and relative isoform abundances for **d** ASD mCherry and **e** ASD Clover. RNA isoform 1-3-4 and its derivatives are colored blue. The RNA isoform abundance percentages include ±1 standard deviation from biological duplicates. mCherry and Clover fluorescence levels for **f** ASD mCherry and **g** ASD Clover with their corresponding spliced controls 1-2-3-4, 1-4, 1-2-4, and 1-3-4. The median fluorescence of each population was measured and normalized to the median fluorescence intensities of the BFP transfection marker and the spliced controls. Median values from biological triplicates were averaged and reported with an error range of ±1 standard deviation. Fluorescence microscopy images of HEK-293T cells transfected with **h** ASD mCherry and **i** ASD Clover. Scale bars represent 50 μm. Source data are provided as a Source Data file

preferential binding of early spliceosome components than the corresponding sequences of exon 2[29]. Given the interplay between these consensus sequences, we developed a graded series of BP and PPT elements to reliably produce alternate isoform 1-2-4. The activities of native BP and PPT sequences were tuned through targeted mutations and truncations (Fig. 3a) and tested in ASD mCherry and ASD Clover. Altering mutually exclusive exon selection in fluorescent ASDs can increase production of one fluorescent signal while reducing the other, thereby enhancing detection of splicing transitions. Consequently, ASDs were first screened for changes in relative fluorescence using flow cytometry and then several were selected for RNA analysis using long-read sequencing.

To favor isoform 1-2-4 production, we first modified the exon 3 BP sequence from GGCTAAC to GGCGATC, deterring exon 3 selection. Normalized fluorescence levels indicate that the modified BP element in ASD mCherry reduces mCherry fluorescence by 33% with a 10% increase in Clover fluorescence relative to the base device (Fig. 3b). 90.8% of the RNA splices into isoform 1-3-4 and 5.5% splices into isoform 1-2-4. Similarly, this modified BP element in ASD Clover generates low levels of mCherry fluorescence (3%) and of RNA isoform 1-2-4 (1.0%) (Supplementary Fig. 5a). Thus, weakening the BP element of exon 3 introduces modest production of alternate isoform 1-2-4.

Another approach for directing selection of exon 2 entails strengthening its BP and PPT sequences. To accomplish this, we

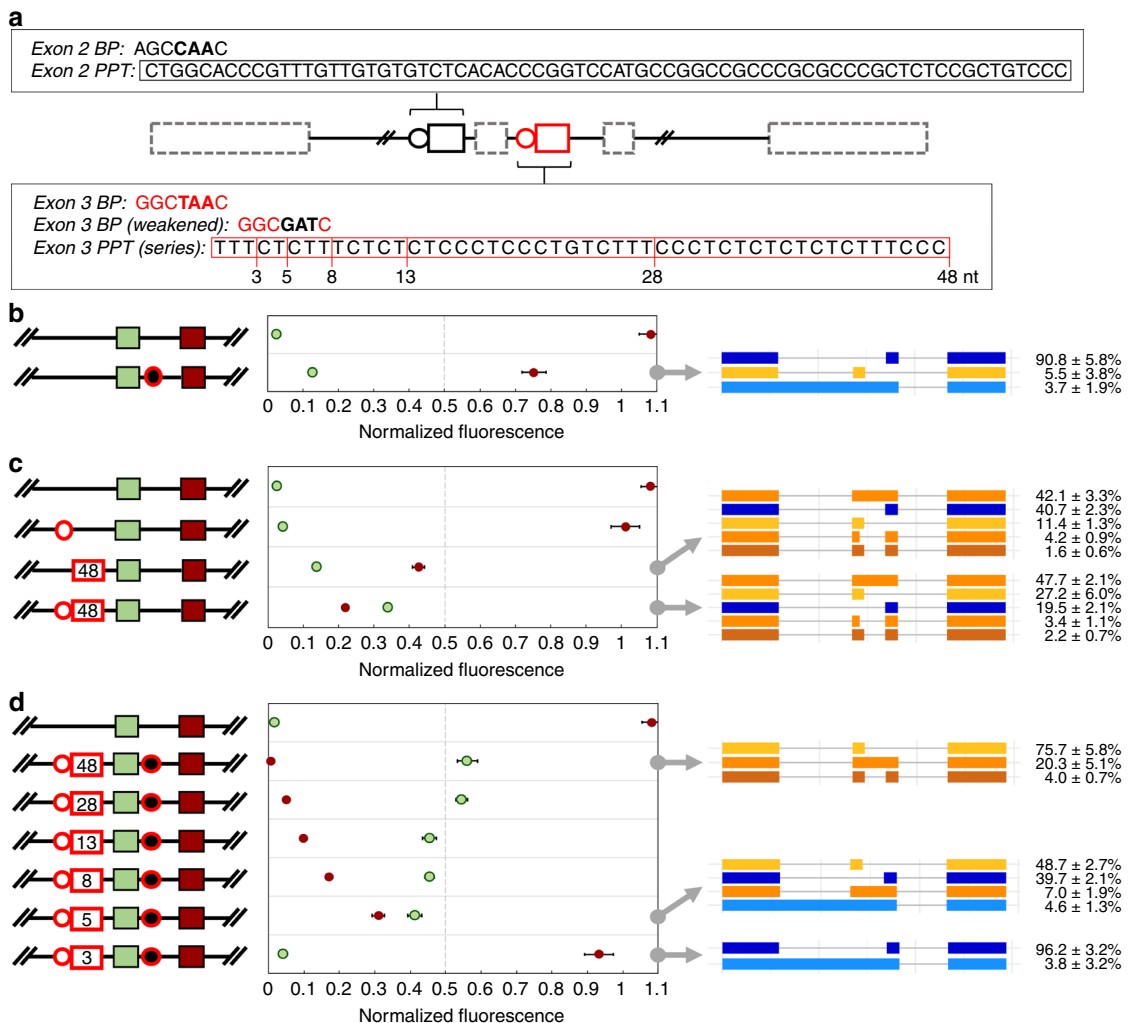

**Fig. 3** Producing alternate isoform profiles by tuning consensus sequence element strengths in the intron framework. **a** Schematic illustrating the BP and PPT elements associated with exon 2 (black) and exon 3 (red). Native and mutated BP and PPT elements are written (bold text denotes the branchpoint adenosine and associated nucleotides; red lines indicate the PPT sequence lengths, in nucleotides, of elements in the exon 3 PPT series). These BP and PPT elements were characterized in ASD mCherry. mCherry and Clover fluorescence levels from ASD mCherry devices containing **b** a mutated exon 3 BP element (black circle with red outline), **c** a mutated exon 2 BP element (white circle with red outline) and a mutated exon 2 PPT element (white rectangle with red outline indicating PPT sequence lengths in nucleotides), and **d** serial truncations of modified exon 2 PPT elements with mutations of the BP elements of both exons 2 and 3. Fluorescence from the splicing devices and controls was quantified via flow cytometry. The median fluorescence of each population was measured and normalized to the median fluorescence intensities of the BFP transfection marker and the spliced controls. Median values from biological triplicates were averaged and reported with an error range of ±1 standard deviation. Devices generating distinctive protein isoform profiles were further assessed using long-read sequencing and the relative abundance of each isoform was quantified. Blue represents RNA isoform 1-3-4 and its derivatives; yellow represents RNA isoform 1-2-4 and its derivatives; and orange represents RNA isoform 1-2-3-4 and its derivatives. The RNA isoform abundance percentages include ±1 standard deviation from biological duplicates. Source data are provided as a Source Data file

substituted these two sequences of exon 2 with the corresponding sequences of exon 3. We mutated the exon 2 BP sequence (AGCCAAC) into the exon 3 BP sequence (GGCTAAC) and replaced the original exon 2 PPT sequence with the 48-nucleotide PPT sequence native to exon 3. We assessed the impacts on splicing of these modified elements in ASD mCherry individually and in combination. Normalized fluorescence data show that mutating the BP element alone minimally alters splicing, with a 7% decrease in mCherry fluorescence and a 2% increase in Clover fluorescence relative to ASD mCherry (Fig. 3c). Modifying only the PPT element produces greater changes, with mCherry fluorescence dropping to 43% and Clover fluorescence increasing to 14%. Altering BP and PPT elements together is synergistic, as mCherry fluorescence lowers to 22% and Clover fluorescence rises to 34%, producing the first ASD mCherry device in which Clover fluorescence surpasses

mCherry fluorescence. Analyzing devices harboring modified PPT elements using long-read sequencing shows significant changes to splicing, with an aberrant double inclusion RNA product being dominant. Of these devices, the secondary RNA product from the device having only the modified PPT element is isoform 1-3-4 (40.7%), while the secondary RNA product from the device harboring both modified BP and PPT elements is isoform 1-2-4 (27.2%). We observe a comparable shift in dominance from isoform 1-3-4 to isoform 1-2-4 upon integrating the modified BP and PPT elements in ASD Clover devices (Supplementary Fig. 5b). Accordingly, strengthening the BP and PPT elements of exon 2 substantially shifts isoform distributions by increasing production of the natively minor isoform 1-2-4.

Based on these results, we hypothesized that BP and PPT elements of both mutually exclusive exons must be tuned

simultaneously to effectively drive isoform 1-2-4 production. Thus, we incorporated three targeted modifications with specific activities into the intron framework. Specifically, (1) the BP element of exon 3 was weakened to the mutated sequence GGCGATC, (2) the BP element of exon 2 was strengthened to the sequence GGCTAAC, and (3) the PPT element of exon 2 was replaced with a 48-, 28-, 13-, 8-, 5-, or 3-nucleotide portion of the PPT sequence native to exon 3. Flow cytometry analysis indicates that a modified ASD mCherry device harboring the 48-nucleotide PPT element expresses 54% Clover fluorescence and 4% mCherry fluorescence (Fig. 3d). When the PPT element is shortened from 48 to 5 nucleotides, the fluorescence levels of Clover and mCherry shift to 41 and 30%, respectively. Further truncating the 5-nucleotide PPT element to 3 nucleotides results in a marked change in isoform levels, with a 39% decrease in Clover fluorescence (to 2%) and a 65% increase in mCherry fluorescence (to 95%). RNA analysis of ASD mCherry devices having the 48-, 5-, and 3-nucleotide PPT elements confirms that isoform profiles correspond to observed fluorescence levels. The device with a 48-nucleotide PPT element dominantly splices to isoform 1-2-4 (75.7%), whereas the device with a 3-nucleotide PPT element primarily splices to isoform 1-3-4 (96.2%). The device with the 5-nucleotide PPT element produces intermediate levels of both isoforms 1-2-4 (48.7%) and 1-3-4 (39.7%). ASD Clover devices harboring the same series of modified BP and PPT elements generate similar isoform profiles, with the transition in dominance of mutually exclusive isoforms occurring between the 13- and 8-nucleotide PPT elements (Supplementary Fig. 5c). Fluorescence microscopy of a subset of ASDs corroborates the observed trends (Supplementary Fig. 6). Comparable splicing outcomes were also seen from several ASDs that were similarly characterized in HeLa, CHO-K1, and U2OS cells (Supplementary Fig. 7). Therefore, a graded series of BP and PPT elements of varying strengths in the intron framework robustly alters RNA isoform profiles and relative fluorescence levels.

**Splicing of transcription factors activates gene expression.** Since MEAS facilitates the exchange of internal exons without disrupting global protein structure, we applied the intron framework to program DNA-binding domains of transcription activator-like effector transcription factors (TALE-TFs). TALE DNA-binding domains consist of repeat monomers, each containing 34 amino acids, of which the two at positions 12 and 13 are termed repeat variable diresidues (RVDs) and confer nucleotide specificity to a DNA target site[30]. A simple code specifies the target nucleotide of each RVD (NG = T, NN = G, NI = A, HD = C).

We assembled two sets of ASDs encoding TALE-TFs (ASD TALE-TFs) with differing DNA-binding domains consisting of 12.5 repeat monomers that recognize 14-nucleotide binding sites, including a 5′ T that is essential for TALE binding. The exons in ASD TALE-TF devices were derived from full-length TALE-TF sequences whose relative activities in activating fluorescent reporters from binding sites harboring one- or two-nucleotide mismatches were previously reported[30,31]. Exonic sequences in ASD TALE-TF devices were designed to encode two monomers within the 138-nucleotide mutually exclusive exons (exons 2 and 3) and remaining monomers within exons 1 and 4 (Fig. 4a). Exon 4 also includes a VP64 transcription activation domain, 2A self-cleaving peptide, and Clover tag. All devices built using the base intron framework were designed to splice by default to isoform 1-3-4.

TALE-TF 1 binds 5′-TACΦACTCACTATA, with binding site variants altering specificity at the underlined Φ. ASD TALE-TF 1 incorporates the third and fourth monomers into both exons 2

and 3, with the programmable third RVD encoding NG, NN, or NI to target nucleotide T, G, or A, respectively (Fig. 4b). TALE-TF 2 binds 5′-TTTTGTΘΘTCTTTA, with binding site variants altering specificity at the underlined dinucleotide ΘΘ. ASD TALE-TF 2 incorporates the sixth and seventh monomers into both exons 2 and 3, with the programmable RVDs encoding NG NG, NN NN, or NI NI to target dinucleotides TT, GG, or AA, respectively (Fig. 4c). We also confirmed that ASD TALE-TF exons contain abundant ESE elements that aid in spliceosome recognition (Supplementary Fig. 8).

We transfected plasmids harboring ASD TALE-TF 1 and ASD TALE-TF 2 devices into HEK-293T cells and analyzed RNA isoforms through long-read sequencing. With ASD TALE-TF 1 variants, we observe at least 78% of spliced RNA are the dominant isoform 1-3-4 (Fig. 4b). Aberrantly-spliced isoforms and unspliced transcripts each account for less than 10% of spliced RNA products. Similarly, isoform 1-3-4 accounts for the majority of spliced RNA produced by ASD TALE-TF 2 variants (Fig. 4c). Three classes of aberrantly-spliced products are generated by ASD TALE-TF devices: (1) isoform 1-3-4 with either intron 3 retained or cryptic splice sites in exons 1 and 4, (2) double exclusion products with cryptic splice sites in exons 1 and 4, and (3) a double exclusion product that incorporates the 3′ ss of exon 2 as the 5′ ss of exon 1 and that contains a cryptic splice site in exon 4. RNA analysis identifies three cryptic splice site positions in ASD TALE-TF 1: 534 nucleotides upstream of the 5′ ss of exon 1, 678 nucleotides downstream of the 3′ ss of exon 4, and 994 nucleotides downstream of the 3′ ss of exon 4. Two cryptic splice site positions are also identified in ASD TALE-TF 2: 412 nucleotides upstream of the 5′ ss of exon 1 and 265 nucleotides downstream of the 3′ ss of exon 4. Despite the cryptic splice sites, our analysis confirms that these ASD TALE-TF devices all dominantly produce RNA isoform 1-3-4 comparably to the fluorescent ASDs.

We subsequently conducted a combinatorial assessment of the functional activity of each ASD TALE-TF and its spliced controls with reporter plasmids harboring each binding site variant. The reporter plasmids contain the binding site upstream of a minimal cytomegalovirus (minCMV) promoter that activates mCherry expression upon TALE-TF binding (Fig. 5a). We co-transfected plasmids encoding an ASD TALE-TF and reporter into HEK-293T cells and assayed them for mCherry activation using flow cytometry. All spliced controls and ASD TALE-TF devices bind to their cognate binding sites, inducing mCherry fluorescence to varying degrees. A greater than eight-fold activation is observed with ASD TALE-TF 1 devices (Fig. 5b), while a greater than two-fold activation is observed with ASD TALE-TF 2 devices (Fig. 5c). ASD TALE-TF 1 devices in which exon 3 encodes RVD NG and ASD TALE-TF 2 devices in which exon 3 encodes RVDs NN NN most strongly activate mCherry fluorescence with a 16-fold and 9-fold induction, respectively, upon binding to their cognate binding sites. We observe crosstalk among ASD TALE-TF 1 devices and reporters, with ~50–75% of maximal reporter activation occurring with non-cognate binding sites. ASD TALE-TF 2 devices exhibit lower off-target binding activity than ASD TALE-TF 1 devices likely due to the two-nucleotide mismatch in the middle of the binding sites of ASD TALE-TF 2 reporters, compared to the single nucleotide mismatch near the beginning of the binding sites of ASD TALE-TF 1 reporters. These results demonstrate that the intron framework supports the programming of complex and repetitive transcription factor devices to activate gene expression.

**Activation of two distinct genes via MEAS of TALE-TF devices.** Programmable ASD TALE-TF devices were designed to produce

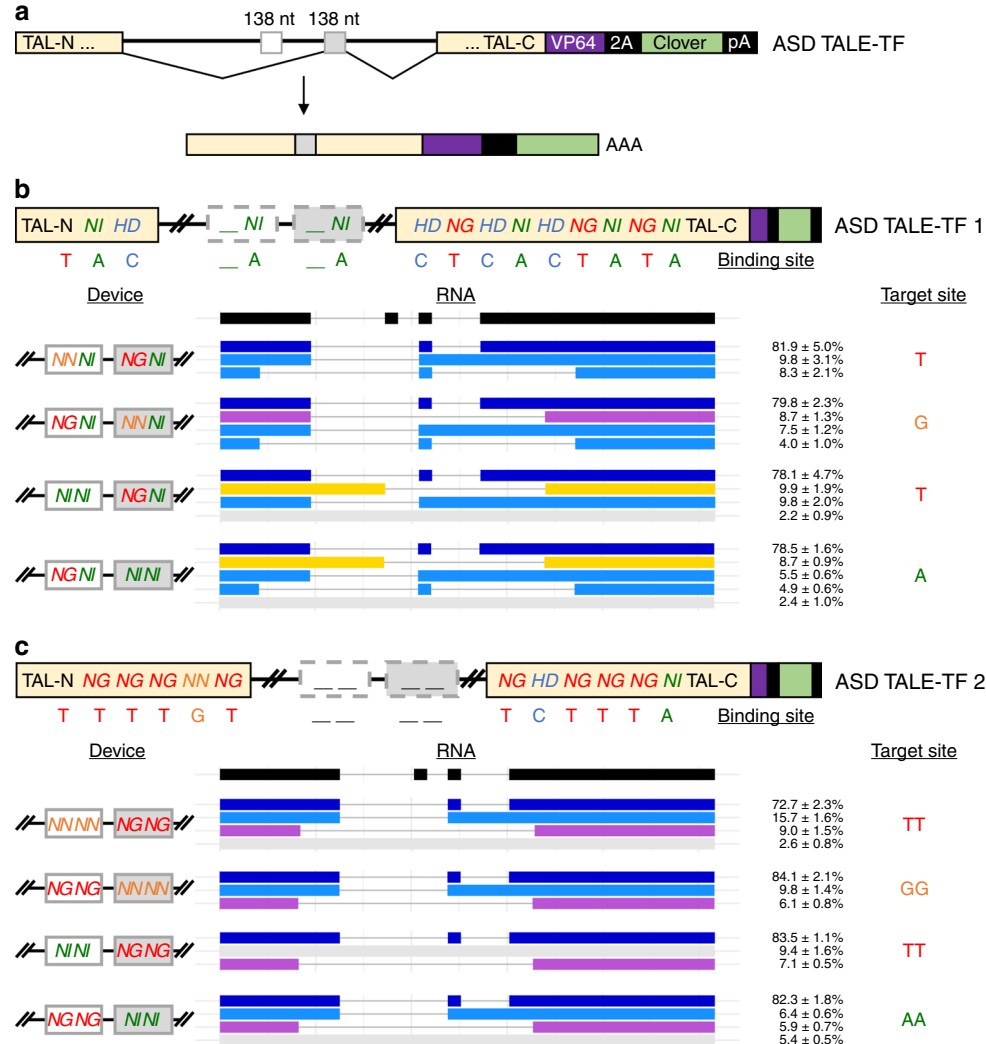

**Fig. 4** Alternatively spliced transcription factors primarily splice to isoform 1-3-4 within diverse RNA isoform profiles. **a** Schematic illustrating an ASD TALE-TF that consists of a modular DNA-binding domain in the mutually exclusive exons. The lengths of the mutually exclusive exons are noted in nucleotides. Exon 4 includes a portion of the DNA-binding domain, a synthetic transcription activation domain (VP64), a self-cleavage peptide (2A), a Clover tag, and a polyA signal (pA). ASD TALE-TF devices built using the base intron framework splice to isoform 1-3-4 by default. **b** A device diagram denotes the composition of ASD TALE-TF 1. ASD TALE-TF 1 incorporates the third and fourth monomers into exons 2 and 3, respectively, with the programmable third RVD encoding NG (red), NN (orange), or NI (green) to target nucleotide T (red), G (orange), or A (green), respectively. RVD HD (blue) is present in exons 1 and 4 and targets nucleotide C (blue). All 1-letter amino acid codes are shown in italicized bold. **c** A device diagram denotes the composition of ASD TALE-TF 2. ASD TALE-TF 2 incorporates the sixth and seventh monomers into exons 2 and 3, respectively, with both RVDs programmed to encode NG NG, NN NN, or NI NI to target dinucleotides TT, GG, or AA, respectively. RNA isoform profiles and relative isoform abundances were characterized using long-read sequencing. Blue represents RNA isoform 1-3-4 and its derivatives; yellow represents RNA isoform 1-2-4 and its derivatives; purple represents RNA isoform 1-4 and its derivatives; and gray represents unspliced transcript. The RNA isoform abundance percentages include ±1 standard deviation from biological triplicates. Source data are provided as a Source Data file

two distinct transcription factors from a single genetic device. Based on our observations with the fluorescent ASDs, we hypothesized that incorporating the graded series of BP and PPT elements into the ASD TALE-TF devices could shift isoform profiles to favor production of either mutually exclusive TALE-TF protein, with each protein binding to a distinct promoter sequence (Fig. 6a). We developed a two-plasmid dual reporter assay with a reporter for TALE-TF isoform 1-2-4 that drives BFP expression and a reporter for TALE-TF isoform 1-3-4 that drives mCherry expression. Each modified ASD TALE-TF was transfected with its two corresponding reporter plasmids into HEK-293T cells, and the spliced RNA and protein activations were assayed.

We constructed a set of ASD TALE-TF 1 devices with the graded series of BP and PPT elements that encode RVD NG in exon 2 (binding nucleotide T) and RVD NN in exon 3 (binding nucleotide G) (Fig. 6b). Devices with these RVDs were selected because they exhibit average gene activations with some crosstalk across binding site variants (Fig. 5b). RNA analysis confirms that devices with the 48-, 28-, and 13-nucleotide PPT elements splice to only one of the two mutually exclusive products, namely isoform 1-2-4 (30.9–35.3%) (Fig. 6b, Supplementary Fig. 9). A device harboring an 8-nucleotide PPT element produces isoform 1-3-4 (48.3%) while still generating isoform 1-2-4 at a reduced level (18.2%). Devices with a 5- or 3-nucleotide PPT element primarily splice to isoform 1-3-4 (52.1%–77.1%). Flow cytometry

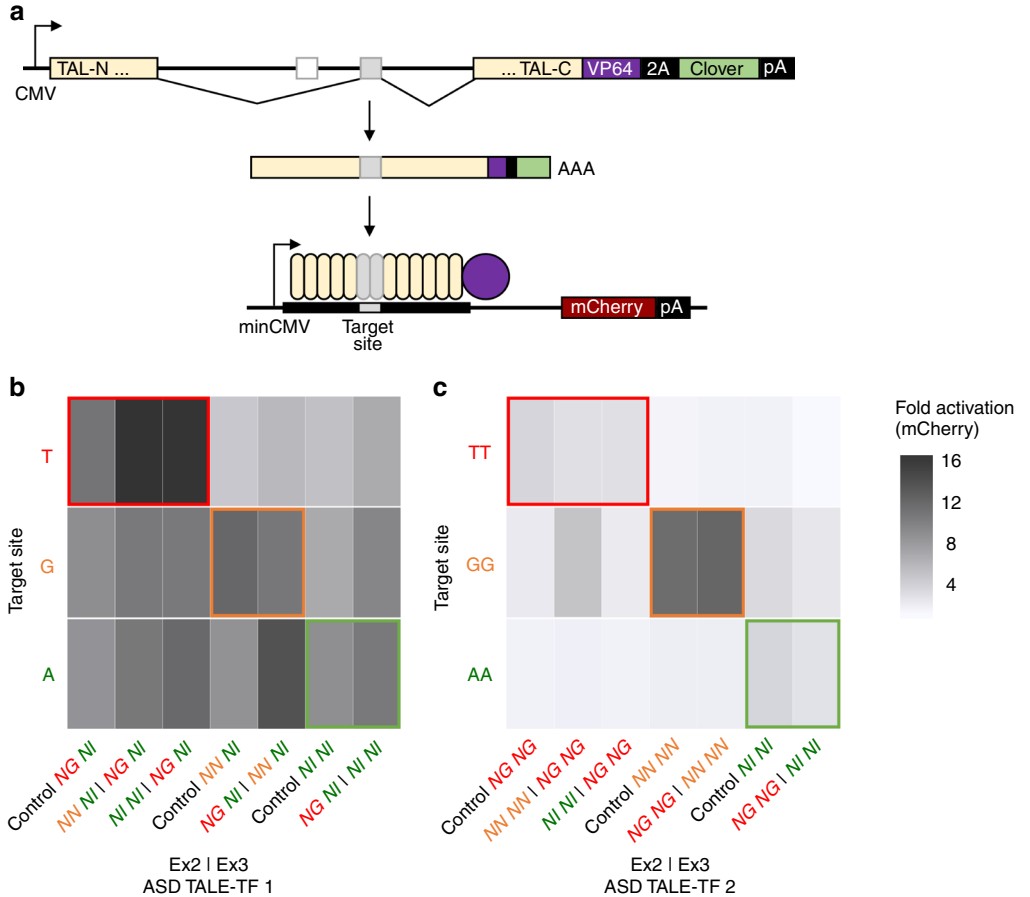

**Fig. 5** Alternatively spliced transcription factors control gene activation from cognate reporters. **a** Schematic representation of the fluorescence reporter system for testing promoter sequence recognition of ASD TALE-TF devices generated by MEAS. A self-cleavage peptide (2A) and Clover tag are included in exon 4 of the ASDs. ASD TALE-TF devices built using the base intron framework splice to isoform 1-3-4 by default to activate mCherry fluorescence. **b** ASD TALE-TF 1 devices and **c** ASD TALE-TF 2 devices were co-transfected with reporters harboring cognate and non-cognate target sites in their binding sites. The fold activation was determined via flow cytometry analysis of mCherry fluorescence in transfected HEK-293T cells, and calculated as the ratio of the median mCherry fluorescence intensity of cells co-transfected with and without the specified ASD TALE-TF. Fold activations from biological triplicates were averaged and reported within an error range of ±1 standard deviation. Colored rectangles (red, orange, and green) indicate activation from cognate reporters. Source data are provided as a Source Data file

analysis of the dual reporter assay corroborates RNA isoform profiles. Production of isoform 1-2-4 corresponding to the T-binding variant activates BFP reporter expression at levels comparable to the spliced T-binding TALE-TF control (~16-fold), whereas isoform 1-3-4 produces the G-binding variant and activates mCherry reporter expression at levels similar to the spliced G-binding TALE-TF control (~10-fold). While some off-target activation is observed, isoforms 1-2-4 and 1-3-4 predominantly activate BFP and mCherry expression from their respective cognate reporters. Fluorescence microscopy of several ASDs corroborates these observed trends (Supplementary Fig. 10). Swapping the fluorescent proteins in the reporters results in analogous splicing outcomes and gene activations (Supplementary Figs. 11a and 11b). We also designed comparable ASD TALE-TF 1 devices targeting T and A binding site variants and observe similar activations in these devices (Supplementary Fig. 12).

Finally, we incorporated the series of BP and PPT elements into ASD TALE-TF 2 devices that encode RVDs NG NG in exon 2 (binding dinucleotide TT) and RVDs NI NI in exon 3 (binding dinucleotide AA) (Fig. 6c). As devices with these RVDs exhibit minimal off-target binding (Fig. 5c), we expected this design would reduce crosstalk between BFP and mCherry activations in our dual reporter assay. RNA analysis reveals that modified ASD TALE-TF 2 devices harboring a 48-, 28-, 13-, and 8-nucleotide PPT element robustly splice to isoform 1-2-4 (58.2%–75.1%) (Fig. 6c, Supplementary Fig. 13). ASD TALE-TF 2 transitions from dominant expression of isoform 1-2-4 to dominant expression of isoform 1-3-4 between the 8- and 5-nucleotide PPT elements, while the same transition occurs in ASD TALE-TF 1 between the 13- and 8-nucleotide PPT elements. The device splices to a maximal level of isoform 1-3-4 (60.9%) with a 3-nucleotide PPT element. Flow cytometry analysis of the dual activation experiments confirms RNA analysis in that activations of BFP (~3-fold) and mCherry (~2-fold) are observed in the presence of their corresponding isoforms. Exchanging the fluorescent proteins in the reporters has minimal impact on the trends in reporter activations (Supplementary Figs. 11a and 11c). We also assessed additional ASD TALE-TF 2 devices targeting TT and GG binding site variants and observed comparable activations (Supplementary Fig. 14). Taken together, these results demonstrate that MEAS of modular transcription factors can effectively activate gene expression from distinct promoter sequences.

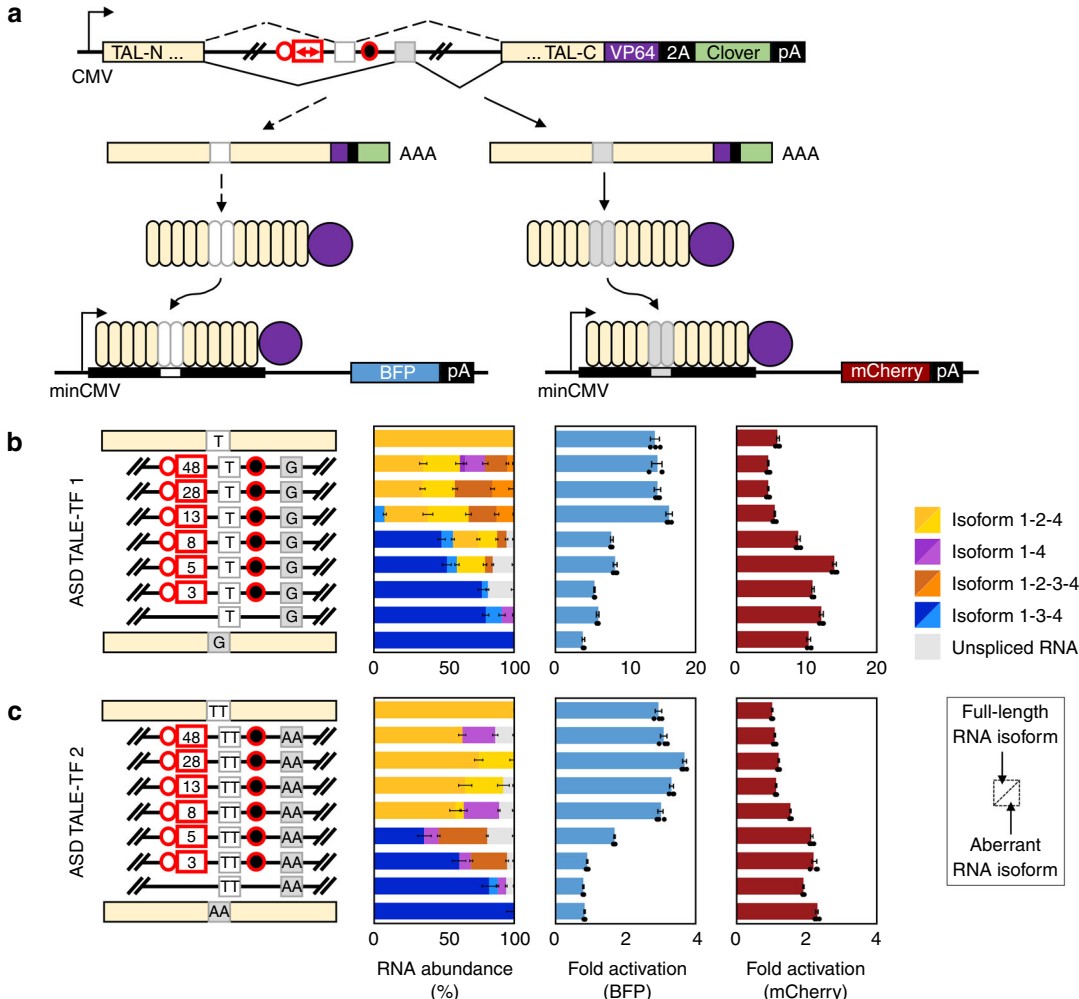

**Fig. 6** Spliced transcription factors generate mutually exclusive isoforms for controlled gene activation from two distinct promoters. **a** Schematic illustrating the activation of two fluorescent proteins from a single MEAS event, where isoform 1-2-4 activates BFP reporter expression and isoform 1-3-4 activates mCherry reporter expression. Devices contain mutated BP elements for exons 2 and 3 (circles with red outlines) and modified exon 2 PPT elements (white rectangle with red outline representing PPT sequence lengths in nucleotides). **b** ASD TALE-TF 1 devices built from a modified intron framework that encode RVD NG (target site T) in exon 2 and RVD NN (target site G) in exon 3. **c** ASD TALE-TF 2 devices built from a modified intron framework that encode RVDs NG NG (target site TT) in exon 2 and RVDs NI NI (target site AA) in exon 3. Relative RNA isoform abundances were characterized using long-read sequencing. Error bars in the RNA isoform data represent ±1 standard deviation from biological triplicates. Blue represents RNA isoform 1-3-4 and its derivatives; yellow represents RNA isoform 1-2-4 and its derivatives; orange represents RNA isoform 1-2-3-4 and its derivatives; purple represents RNA isoform 1-4 and its derivatives; and gray represents unspliced transcript. The fold activation was determined via flow cytometry analysis of BFP and mCherry fluorescence in transfected HEK-293T cells, and calculated as the ratio of the median BFP or mCherry fluorescence intensity of cells co-transfected with and without the specified ASD TALE-TF. Fold activations from biological triplicates were averaged and reported within an error range of ±1 standard deviation. Source data are provided as a Source Data file

## Discussion

Alternative splicing-based approaches provide a powerful strategy for expanding coding capacity and for producing diverse RNA and protein isoforms from a single genetic device. Despite advances in understanding of the splicing code[32], broader application of alternative splicing in engineered biological systems has been hindered by the interdependencies of sequence elements that extend across exonic and intronic sequence space. By developing design rules for MEAS, we establish an intron framework that supports the splicing of modular and programmable exonic sequences. Proximity of the BP sequence of exon 3 to the 5′ ss of exon 2 enforces mutual exclusivity in the intron framework. We also describe a graded series of BP and PPT elements for co-expressing mutually exclusive RNA and protein isoforms at varying static ratios. Since 80% of the 1,339 predicted mutually exclusive exons in the human genome are arranged in pairs[33], this

three-intron four-exon platform captures a primary presentation of natural MEAS. By producing ASDs encoding functionally-distinct fluorescent proteins and modular transcription factors, we demonstrate the broad versatility and extensibility of this platform.

The α-tropomyosin minigene from which we harvested the genetic components for our intron framework contains mutually exclusive exons that are 126 nucleotides long. We determined that mutually exclusive exons ranging in length from 117 to 150 nucleotides are tolerated in ASD mCherry, ASD Clover, and ASD TALE-TF devices, thus establishing a widened permissible range of mutually exclusive exon lengths. The majority of alternatively spliced exons in the human genome range from 50 to 200 nucleotides[34,35], so we expect that the intron framework can accept mutually exclusive exons with lengths that are outside of the range that we tested. Exons shorter than 50 nucleotides may

impose a requirement for an ESE[36,37]. Constraints based on maximum length are less clear[38] and likely to be sequence-specific, as cryptic splice sites may arise in longer exonic sequences to create pseudo-exons in the intron framework.

Our studies indicate that conservation of splice sites at exon–intron junctions is important for accurate MEAS, with some sequence flexibility. Conserving splice sites is generally feasible through rational design, particularly in cases where the amino acid at the exon boundary is permissive to mutation. To conserve the native 3′ ss of exon 4 in ASD mCherry, we mutated a leucine codon to an alanine codon. This change is well tolerated, in that mCherry and Clover can be expressed by the device. However, mutation of a glycine codon to an alanine codon in ASD Clover at the same exon–intron junction abolishes mCherry fluorescence. While this 3′ ss in ASD Clover conforms to the consensus sequence[39], the efficiency of accurate exon 4 selection decreases due to the appearance of a cryptic splice site.

Cryptic splice sites typically result in reduced levels of desired RNA isoforms in favor of aberrant products. Aberrant products can also result from intron inclusion when defined splice sites are overlooked. We speculate that aberrant products arise because exon sequence composition impacts relative splice site strength. We found that predicting cryptic splice sites a priori in programmable exons is nontrivial given the combinatorial nature of splicing regulation. Early optimizations geared towards effective reduction of cryptic splice sites could result in unanticipated downstream effects, such as the emergence of novel cryptic splice sites. Future high-throughput analyses of splicing patterns of synthetic genes[40] may reveal sequence constraints and mechanistic insights to enable improved prediction of and removal of cryptic splice sites in ASDs. Despite the variable exon sequences across ASDs, a large fraction of the transcripts generated by these devices splice to the intended mutually exclusive isoforms.

By engineering a graded series of BP and PPT elements entirely within the static regions of the intron framework, we significantly alter splicing outcomes using a relatively small number of rationally-designed devices rather than a library screening approach. The dominant isoform shifts from 1-3-4 to 1-2-4 when the BP and PPT elements in intron 1 are strengthened and the BP element in intron 2 is weakened. Splicing profiles can be reverted back to dominantly producing isoform 1-3-4 by truncating the modified PPT element in intron 1 to a shortened 3- to 8-nucleotide PPT element, which indicates a critical PPT nucleotide threshold for mutually exclusive exon selection. These results mirror observations in endogenous genes where BP and PPT sequences can drive binding kinetics that direct spliceosome assembly. In the α-tropomyosin minigene, the splicing factor U2AF65 and the polypyrimidine tract-binding protein (PTB) preferentially interact with the BP and PPT sequences in intron 2 early in spliceosome assembly[29,41] to generate isoform 1-3-4. By introducing this graded series of BP and PPT elements into intron 1, we shift the splicing patterns to favor isoform 1-2-4 and confirm that these elements guide early spliceosome assembly. While subtler in effect than BP and PPT elements, engineered intronic splicing enhancers (ISE) and intronic splicing silencers (ISS) may provide an additional layer of control over spliced isoforms.

We demonstrate that this MEAS platform enables compression of a protein coding sequence by approximately 38% from 16 kilobases to 10 kilobases by expressing two functionally-distinct TALE-TFs from a single genetic device. Notable cases of natural MEAS, such as in the *Dscam* gene in *Drosophila melanogaster* where 95 alternative exons in four-exon clusters generate 38,016 isoforms[42,43], motivate the potential for extending the MEAS platform to further increase genomic efficiency and incorporate dynamic control. MEAS platforms like this one could be advantageous for programming diverse sets of protein isoforms and

tuning their ratios in many applications including chimeric antigen receptors[44], localization tags[45,46], antibody variable domains[47], and enzyme catalytic active sites[48]. Such MEAS platforms could also be adopted to control alternative splicing in plants[49] and flies[50], as the general mechanism of splicing and many of the proteins involved are conserved across eukaryotes. Ultimately, the ability to systematically program alternative splicing will enable the construction of sophisticated genetic devices that expand RNA and protein diversity to advance the scale and complexity of engineered biological systems.

## Methods

**Design and construction of fluorescent ASDs.** All plasmids were constructed using modern molecular biology techniques. Oligonucleotide synthesis was performed by Integrated DNA Technologies, Inc. (Coralville, IA) and Stanford Protein and Nucleic Acid Facility (Stanford, CA). Enzymes used for cloning, including restriction enzymes and T4 DNA ligase, were acquired from New England Biolabs (Ipswich, MA). DNA polymerases were obtained from Agilent Technologies (Santa Clara, CA). DNA products were transformed into chemically-competent *Escherichia coli* strain TOP10 (Invitrogen, Carlsbad, CA) by heat shock and clones were selected on LB agar plates with 50 mg/L ampicillin. Clones were then verified through colony polymerase chain reaction (PCR) and grown in LB liquid media with ampicillin. Plasmids were prepared from *E. coli* using Econospin columns (Epoch Life Science, Missouri City, TX) or Wizard Plus SV Miniprep DNA Purification systems (Promega, Madison, WI) according to manufacturer's instructions. Plasmids were sequence verified by Elim Biopharmaceuticals, Inc. (Hayward, CA).

All of the fluorescent ASDs and their associated controls were cloned into the pCS2585 backbone (Supplementary Table 1). pCS2585 was modified from the pcDNA5/FRT (pCS218) backbone (Invitrogen, Carlsbad, CA) with the integration of an expression cassette encoding EF1a-pTagBFP-HSVtk polyA signal at the BglII restriction site. The expression cassette was assembled using the primers listed in Supplementary Table 2.

ASD mCherry and ASD Clover were constructed using homologous recombination with a modified GeneArt High-Order Genetic Assembly protocol (Thermo Fisher Scientific, Waltham, MA). Three-intron sequences (Supplementary Table 3) and four-exon sequences (Supplementary Table 4) for each ASD were amplified from their corresponding plasmids by PCR using PfuUltraII Fusion HS DNA Polymerase (Agilent Technologies, Santa Clara, CA) with primers (Supplementary Table 2) that share end-terminal homology for assembly into the linearized pYES1L vector. A positive clone for each ASD was amplified from the pYES1L vector by PCR and subcloned into pCS2585 between the BamHI and NotI restriction sites using Gibson assembly. The ASDs harboring BP and PPT element modifications were constructed using primers (Supplementary Table 2) for the QuikChange Site-Directed Mutagenesis Kit (Agilent, Santa Clara, CA) and Gibson assembly, and then directly cloned into pCS2585. All plasmids are listed in Supplementary Table 1, and plasmid maps for pCS2585, pCS3708 (ASD mCherry), and pCS3736 (ASD Clover) are shown in Supplementary Fig. 15.

**Bioinformatic analysis of ESE and ESS motifs.** Exonic splicing enhancer and silencer sequences in the mutually exclusive exons were identified using the Human Splicing Finder (HSF v3.1)[51], a bioinformatics tool (http://www.umd.be/HSF3/). The software identifies *cis*-acting regulatory motifs using published algorithms, such as RESCUE- ESE[52] and ESE-Finder[53], as well as new algorithms designed to use available or newly created matrices.

**Mammalian cell culture.** HEK-293T cells (ATCC, Manassas, VA), HeLa cells (a generous gift from the James Chen Laboratory, Stanford, CA), CHO-K1 cells (ATCC, Manassas, VA), and U2OS cells (a generous gift from the Katrin Chua Laboratory, Stanford, CA) were cultured in DMEM media (Thermo Fisher Scientific, Waltham, MA) supplemented with 10% FBS (Thermo Fisher Scientific, Waltham, MA) and passaged regularly. All cells were grown at 37 °C, 5% $CO_2$, and 80% humidity in an incubator. All cells were acquired from trusted sources and visually authenticated using a Zeiss Axiovert 200 M Inverted Microscope (ZEISS, Oberkochen, Germany). Good aseptic technique was employed to reduce the risk of mycoplasma contamination.

**Transient transfections.** HEK-293T, HeLa, CHO-K1, and U2OS cells were transiently transfected with an ASD or a spliced control using Lipofectamine 2000 (Thermo Fisher Scientific, Waltham, MA) according to manufacturer's instructions. HEK-293T, HeLa, and U2OS cells were seeded at 50,000 cells/well while CHO-K1 cells were seeded at 40,000 cells/well in 24-well plates 24 h prior to transfection. Each transfection sample received 500 ng of plasmid DNA at a 3:1 ratio of Lipofectamine 2000 to DNA.

**Long-read sequencing library preparation**. Libraries were prepared according to Pacific Biosciences' Iso-Seq method and spliced isoforms of several ASDs were multiplexed using barcode sequences. RNA was extracted using the GenElute Mammalian Total RNA Miniprep Kit (Millipore Sigma, St. Louis, MO) according to manufacturer's instructions. Reverse transcription of the spliced isoforms for each device was performed separately in a total volume of 10 μl with 1 ug of total RNA using SuperScript III reverse transcriptase (Invitrogen, Carlsbad, CA) with the bGH R primer (Supplementary Table 2) according to manufacturer's instructions. The single-stranded cDNA was amplified in large-scale PCR reactions using a primer pair targeting the first and last exons of the spliced isoforms for 20 PCR cycles with KAPA DNA Polymerase (Roche, Basel, Switzerland) according to manufacturer's instructions. The primer pair encodes forward and reverse barcodes for amplicon sequencing provided by Pacific Biosciences. Large-scale PCR products were purified with AMPure PB beads (Pacific Biosciences, Menlo Park, CA) according to manufacturer's instructions and quantified using Qubit HS dsDNA Kit (Thermo Fisher Scientific, Waltham, MA), for which quality control was performed using the High Sensitivity DNA Kit on a 2100 Bioanalyzer (Agilent Technologies, Santa Clara, CA). Equimolar ratios of PCR products were pooled together to create libraries that were prepared using the SMRTbell Template Prep Kit 1.0 (Pacific Biosciences, Menlo Park, CA) according to manufacturer's instructions.

Libraries were primed for sequencing by annealing a sequencing primer (component of the SMRTbell Template Prep Kit 1.0) and then binding polymerase from the DNA/Polymerase Binding Kit P6 (Pacific Biosciences, Menlo Park, CA) to the primer-annealed template, according to manufacturer's instructions. The polymerase-bound template was bound to magnetic beads from the MagBead Kit v2 (Pacific Biosciences, Menlo Park, CA) and sequenced on the PacBio RS II sequencer (Pacific Biosciences, Menlo Park, CA). Each device was run on two or three cells from a SMRT Cell 8Pac v3 (Pacific Biosciences, Menlo Park, CA) and 360 min movies were collected.

**Long-read sequencing data analysis**. Subread filtering was performed using SMRT analysis software (v2.3.0) (Pacific Biosciences, Menlo Park, CA). Full-length read information was gathered using the RS_ReadsOfInsert.1 protocol. The reads corresponding to spliced isoforms from each device were demultiplexed, using Pacific Biosciences' protocols and tutorials (https://github.com/PacificBiosciences/cDNA_primer/), and classified into CCS (Circular Consensus Sequences) and non-CCS subreads by ToFu[54]. The isoform-level clustering algorithm ICE (Iterative Clustering for Error Correction) was run and the results were polished using Quiver. Transcripts lacking the 5′ end of exon 1 or the 3′ end of exon 4 were removed. Reads were mapped to the ASD to generate spliced isoform profiles and relative abundance estimates. Only transcripts accounting for 1% or more of the total number of reads are reported. The number of reads and the sequence corresponding to each spliced isoform generated by an ASD is available in the Supplementary Data 1 file.

**Fluorescence quantification and data analysis of ASDs**. Fluorescence data were obtained 48 h after transfection using a MACSQuant VYB flow cytometer (Miltenyi Biotec, Bergisch Gladbach, Germany) equipped with 405 nm, 488 nm, and 561 nm lasers. mCherry, Clover, and BFP were measured through 615/20 nm, 525/50 nm, and 450/50 nm band pass filters, respectively.

Flow cytometry data were analyzed using FlowJo 7 software (Tree Star, Ashland, OR). Minimal compensation was required as the three colors are orthogonal. Viability was gated by side scatter and forward scatter followed by gating for singlets. Singlets were further gated for BFP-positive cells, where BFP served as an internal control for transfection efficiency, and then mCherry- or Clover-positive populations. Representative flow cytometry plots and the gating strategy are shown in Supplementary Fig. 16. Median fluorescence values from each device were used to calculate mCherry and Clover fluorescence percentages as follows:

$$\%\text{mCherry in ASD mCherry} = \frac{\frac{\text{Sample mCherry}}{\text{Sample BFP}}}{\frac{\text{Spliced isoform } 1-3-4 \text{ control mCherry}}{\text{Spliced isoform } 1-3-4 \text{ control BFP}}} \quad (1)$$

$$\%\text{Clover in ASD mCherry} = \frac{\frac{\text{Sample Clover}}{\text{Sample BFP}}}{\frac{\text{Spliced isoform } 1-2-4 \text{ control Clover}}{\text{Spliced isoform } 1-2-4 \text{ control BFP}}} \quad (2)$$

$$\%\text{Clover in ASD Clover} = \frac{\frac{\text{Sample Clover}}{\text{Sample BFP}}}{\frac{\text{Spliced isoform } 1-3-4 \text{ control Clover}}{\text{Spliced isoform } 1-3-4 \text{ control BFP}}} \quad (3)$$

$$\%\text{mCherry in ASD Clover} = \frac{\frac{\text{Sample mCherry}}{\text{Sample BFP}}}{\frac{\text{Spliced isoform } 1-2-4 \text{ control mCherry}}{\text{Spliced isoform } 1-2-4 \text{ control BFP}}} \quad (4)$$

The "sample" represents the device and the "spliced isoform control" represents the isoform that would be generated upon splicing and that exemplifies the maximum expected mCherry or Clover fluorescence from a device. Median

fluorescence values from biological triplicates were averaged and reported with an error range of ±1 standard deviation.

**Fluorescence microscopy**. Fluorescence microscopy was performed on an Upright Zeiss AxioImager Epifluorescence/Widefield Microscope (ZEISS, Oberkochen, Germany) equipped with a Zeiss Axiocam 503 mono camera (ZEISS, Oberkochen, Germany) and EXFO X-Cite 120 (Excelitas Technologies, Fremont, CA). Zeiss Zen pro software (ZEISS, Oberkochen, Germany) was used to set up fluorescence imaging. Images from pTagBFP (filter cube DAPI, Semrock Brightline #1160 A) (Semrock, Rochester, NY), mCherry (filter cube TXRED, Semrock Brightline #4040B), and Clover (filter cube GFP, Semrock Brightline #3035B) channels were taken 48 h post-transfection with a 20×/0.8 M27 Plan-Apo objective. Image processing was performed in Fiji (ImageJ).

**Design and construction of ASD TALE-TF devices and reporters**. ASD TALE-TF 1 and ASD TALE-TF 2 were designed based on validated TALE activators TALE-TF 1 (dTALE1) and TALE-TF 2 (dTALE13)[31]. TALE-TF 1 and TALE-TF 2 bind their cognate sites on minCMV-mCherry reporters and were previously tested with reporters containing one or two base pair mismatches[31]. Each full-length TALE-TF was divided into three fragments for integration into the intron framework, including: (1) a conserved region containing the TAL N-terminus and constant TAL monomers (exon 1), (2) a variable region containing two TAL monomers (exons 2 and 3), and (3) a conserved region containing constant TAL monomers, the TAL C-terminus, a synthetic transcription activation domain (VP64), a self-cleavage peptide (2A), a Clover tag, and a polyA signal (pA) (exon 4). For ASD TALE-TF 1, repeat 3 is the variable monomer. Portions of repeats 3 and 4 are combined to maintain a mutually exclusive exon length of 138 nucleotides. For ASD TALE-TF 2, repeats 6 and 7 are the variable monomers.

This cloning strategy mimics the established hierarchical ligation cloning protocol for assembling full-length designer TALE-TFs. The ASD TALE-TF devices were constructed using a two-step cloning protocol with previously constructed TAL monomers and destination vectors[31] as well as the primers listed in Supplementary Table 2. These TAL monomers contain fewer repetitive sequences to minimize homology and reduce recombination.

In the first step of this cloning strategy, each exonic and intronic portion of the three fragments was amplified from individual TAL monomers and intron sequences (Supplementary Table 3) and assembled using hierarchical ligation with Type IIs BsaI restriction sites, followed by Golden Gate assembly of each fragment with terminal BsmBI restriction sites into the pLenti-EF1a-Backbone(NI) vector (pCS3809) (Supplementary Table 1). Care was taken to ensure that all of the fragments were of roughly equal lengths to make a simple one-pot Golden Gate assembly reaction into the pLenti destination vector. In fragment 1, at the 5′ ss of exon 1, the terminal conserved TAL repeat was amplified to contain only the amino acids TPEQ. In fragment 3, at the 3′ ss of exon 4, the conserved TAL repeat was amplified starting at the amino acids ALE of the previous TAL monomer. All BsaI and BsmBI overhangs of these two fragments were adapted to ensure simple cloning into the pLenti destination vector and to maintain all exon–intron junctions. In fragment 2, the mutually exclusive exons were amplified to begin with the amino acids VVAIAS(RVD), where (RVD) signifies two amino acids that confer specificity to the target site, and to terminate with the amino acids GGKQ of the second monomer, so that splicing of a mutually exclusive exon with exons 1 and 4 forms the complete monomers. Fragment 2, which contains the two mutually exclusive exons, intron 2, and overhangs for introns 1 and 3, was not cloned into the pLenti destination vector because fragment 2 does not require the TAL N- and C-termini. However, fragment 2 did contain BsmBI overhangs and was assembled via Golden Gate assembly.

In the second step of this cloning strategy, each of the three fragments was amplified using unique overhangs provided by the pCS218 backbone at the BamHI and NotI restriction sites and by the intron framework. Primers for fragments 1 and 3 amplify from the TAL termini and the intron sequences in order to avoid primer binding in the repetitive TAL monomers. Primers for fragment 2 amplify from the intron 1 and intron 3 overhangs. Gibson assembly of the three fragments was performed for 15 min using 0.02 pmol of each fragment and was transformed with linearized pCS218 into STBL3 cells (Invitrogen, Carlsbad, CA). All of the TALE exon sequences from the assembled ASD TALE-TF devices are available in Supplementary Table 4. The ASD TALE-TF devices harboring BP and PPT element modifications were constructed using primers (Supplementary Table 2) for QuikChange site-directed mutagenesis and Gibson assembly, and then were directly cloned into pCS218. All plasmids are listed in Supplementary Table 1, and a plasmid map for ASD TALE-TF 1 T/G (pCS3825) is available in Supplementary Fig. 15.

The binding site sequences for TALE-TF 1 and TALE-TF 2 were cloned into the reporter plasmids with mCherry or BFP between the XbaI and BamHI restriction sites. Primers listed in Supplementary Table 2 were annealed with overhangs corresponding to the two restriction sites and introduced into the reporter plasmid using Gibson assembly. All reporter plasmids are listed in Supplementary Table 1.

**Reporter activation assay and data analysis**. HEK-293T transient co-transfections of the ASD TALE-TF devices or controls with reporter plasmids were

performed using Lipofectamine 2000 according to manufacturer's instructions. Seven hundred nanograms of the ASD TALE-TF plasmid was co-transfected with 100 ng of the reporter plasmid. To measure the basal activity of each reporter plasmid, 100 ng of the reporter plasmid was co-transfected with 700 ng of pCS218, which does not express any fluorescent protein. For dual activation studies, 50 ng of each reporter plasmid was co-transfected for the assay. Cells were seeded at 100,000 cells/well in 24-well plates 24 h prior to transfection. Each transfection sample received 800 ng total of plasmid DNA at a Lipofectamine 2000:DNA ratio of 0.4:1. Fluorescence data were obtained 48 h after transfection using a MACSQuant VYB flow cytometer.

Flow cytometry data were analyzed using the FlowJo 7 software. Viability was gated by side scatter and forward scatter followed by gating for singlets, and then mCherry- or BFP-positive populations were gated for activation studies. Representative flow cytometry plots and the gating strategy are shown in Supplementary Fig. 17. Median fluorescence values were used to calculate fold activation as follows:

$$\text{Fold activation} = \frac{\text{mCherry/BFP from TALE} + \text{BS}}{\text{mCherry/BFP from BS}} \qquad (5)$$

"mCherry/BFP" represents the median mCherry or BFP fluorescence intensity based on which color was being activated. Fold activations from biological triplicates were averaged and reported with an error range of ±1 standard deviation.

**Statistical analysis**. All data are presented as means ± 1 standard deviation. Replicate values are in triplicate unless indicated otherwise.

**Reporting summary**. Further information on research design is available in the Nature Research Reporting Summary linked to this article.

## Data availability

The authors declare that all data supporting the findings of this study are available within the paper and its supplementary information files or from the authors upon reasonable request. Source data underlying Figs. 2d–g, 3b–d, 4b–c, 5b–c, 6b–c, and Supplementary Figs. 1b, c, 2b, c, 4b, c, 5a–c, 7a–c, 9a–f, 11b, c, 12a, b, 13a–f, and 14a, b are provided in the Source Data file. Long-read sequencing data from the splicing devices are available in the Supplementary Data 1 file.

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

## Acknowledgements

We thank Dr. Benjamin Kotopka for valuable comments on the manuscript. We thank Dr. Christopher W. J. Smith for sharing the TS23D plasmid. We are grateful to Dr. Joseph Puglisi for allowing us to conduct long-read sequencing experiments on his laboratory's PacBio RS II sequencer and Dr. Elizabeth Tseng for her guidance with long-read sequencing data analysis. Fluorescence microscopy was performed at the Stanford Cell Sciences Imaging Facility. This work was supported by the National Institutes of Health, National Science Foundation, Human Frontiers Science Program (grants to C.D. S.), Stanford Bio-X Institute, National Science Foundation, Siebel Scholars Foundation (graduate student fellowships to M.M.), Stanford Enhancing Diversity through Graduate Education in STEM, Agilent Graduate Fellowship, and the National Institute of General Medical Sciences of the National Institutes of Health [award number T32GM008412] (graduate student fellowships to C.M.K.). Certain commercial equipment, instruments, or materials are identified in this paper in order to specify the experimental procedure adequately. Such identification is not intended to imply recommendation or endorsement by the National Institute of Standards and Technology, nor is it intended to imply that the materials or equipment identified are necessarily the best available for the purpose.

## Author contributions

M.M. and C.D.S. conceived of the project and designed the experiments. M.M., C.M.K, and C.D.S. wrote the manuscript. M.M., C.M.K., S.S.R. and E.M.S. constructed the alternative splicing devices, performed the flow cytometry experiments, and analyzed the results. M.M. performed the long-read sequencing experiments, and M.M. and S.A.M. analyzed the long-read sequencing results. All authors read and commented on the manuscript.

## Additional information

**Competing interests:** M.M. and C.D.S. are inventors on an issued patent (US 10,053,697) that was assigned to Stanford University and relates to the mutually exclusive production of splicing products from an RNA. The remaining authors declare no competing interests.

