## [Peer Review File · Nature Communications]

Reviewers' Comments:

Reviewer #1:

Remarks to the Author:

Overall this is an excellent and timely paper. No new experiments are recommended.

I recommend acceptance after some changes in the writing to make the manuscript easier to understand and the conclusions more in evidence.

The paper fills an important gap in mammalian synthetic biology: systematic engineering of alternative splicing for the expression of protein variants and artificial genetic circuits.

The paper also describes, possibly for the first time, a toolkit consisting of a graded series of splice acceptors of different strengths, such that the ratio of two spliced products can be chosen by the designer. The paper also describes methods for making such a graded series that involve making only a few constructs instead of a large library. This is quite useful and deserves a mention in the abstract.

To get their MEAS system to work, the authors appear to rely on two essential tricks: (1) placement of the alternatively spliced elements so close to each other that the intron between them is too small to allow splicing – thus enforcing mutual exclusivity; and (2) varying the strengths of the splice acceptors upstream of the alternatively spliced exons, resulting in an intermediate level of use and thus in co-expression of both protein variants. This should be clearly stated in the Abstract and elsewhere (perhaps at the beginning of the Results and/or Discussion). As it is, the term “framework” is used but there is no concise description of what the framework is.

The manuscript is generally well written.

Major Recommendations:

1. It is hard to look at Figure 2a and understand how expression of two different fluorescent proteins can result. Where are the start codons and stop codons for each ORF? The figure is particularly confusing because the ‘Clover’ and ‘mCherry’ mRNA isoforms have a significant amount of the other color. (Apologies if I missed this in the Supplementary Information, but the main text and figures should be self-sufficient for things like this.)
2. The manuscript describes a significant amount of data from sequencing of mRNA pools. The description of these data generally seems to assume that the relative abundance of these mRNAs results from different levels of splicing. However, an alternative explanation is that nonsense-mediated decay (e.g. see Lejeune BMB Rep. 2017; 50(4): 175-185) influences the relative levels of mRNAs. The experiments with the TALENs are designed in such a way that this is not an issue, but for the fluorescent proteins this could be an issue, at least based on the way the mRNAs are diagrammed - it is important to know where the start at stop codons are for each ORF relative to the splice sites to evaluate this issue. To be clear, I do not recommend that experiments be done to measure nonsense-mediated decay directly. This would be prohibitive and would also expand the manuscript beyond a single publishable unit.

Minor recommendations

3. In line 102, the phrase “Second, to avoid frameshift mutations” should be changed to “Second, to avoid splicing events that result in frameshifts” (A mutation is a heritable change in DNA.)
4. Use the word “use” instead of “utilize” throughout, just to simplify sentences that are inevitably using large word. For example, in the second sentence of the abstract, replace “underutilized” with “underused”.
5. In the Discussion (line 435-7), the authors describe a set of situations where engineered alternative splicing could be used. Voges et al. (Journal of Biological Engineering 2013, 7:20) have already used engineered alternative splicing for diversification of localization tags. The authors

could cite this reference and try to find other examples where alternative splicing has been used in synthetic biology (which may be rare or nonexistent), and possibly rephrase the sentence to describe why MEAS would be particularly valuable in certain situations.

6. The authors should include some flow cytometry plots in their figures.

7. In line 261, the authors could use the 3-letter code for amino acids to more clearly distinguish amino acids and nucleic acid bases, especially since some 1-letter codes for amino acids may also refer to variable bases.

Reviewer #2:

Remarks to the Author:

In this manuscript, Mathur et al. engineered an intron framework, which they claim can be used in creating programmable devices to preferentially select variable isoforms from mutually exclusive splicing events. The intron framework is constructed based on a 4-exon-3-intron rat α -tropomyosin minigene system in which exons 2 and 3 are mutually exclusive by the steric restraint in spliceosome assembly between them. By splitting the coding regions of fluorescent proteins—mCherry and Clover—as exons, they demonstrate that by altering intronic elements, they can change the amount of each mutually exclusive exon that is incorporated in the final transcripts and eventually in the coded proteins. The authors show that in the intron framework used, 1-3-4 is the predominant isoform and this is due to a well-defined branch point (BP) and polypyrimidine tract (PPT) sequence in intron 2 downstream of exon 2 leading to skipping of exon 2 and inclusion of exon 3. They also show that moving this BP and PPT from intron 2 to intron 1, as well mutating the BP of intron 2, produces 1-2-4 as the predominant isoform, along with dependence of this phenomena on the length of the PPT sequence. Finally, as a proof of concept, the authors generated a TALE transcription factor with mutually exclusive exons coding for variable DNA binding domains and showed that by switching cassettes, the transcription factor could induce expression of reporter genes carrying different DNA binding elements.

Overall, the study is scientifically sound and well executed, but at this stage contrary to author's claims, their MEAS system is not truly programmable. For instance, while their intron framework is tunable to support different ratios of expression of the two isoforms, these ratios are static depending on the specific framework utilized. It would be much more exciting if the ratios were dynamically controlled in response to intrinsic or extrinsic cues. Furthermore, designing other constructs will not be trivial as they will likely include exonic contexts which may shift the splicing ratios in unexpected ways. It is difficult to envision how these devices may be used in practical applications.

Major Comments:

1. The authors claim that their MEAS/ASD system allows an easy way to swap out mutually exclusive regions or domains of a protein to alter function. However, what is the advantage of using this framework as opposed to simply creating separate constructs containing the sequences of interest and bypassing splicing altogether? Doing this would completely reduce aberrantly spliced or unspliced isoforms. Currently, it is unclear as to why this platform would be advantageous over the straight forward old-school method of expressing clones of individual isoforms. Although it would make sense to design such a device if the two mutually exclusive isoforms were to be expressed simultaneously in a cell in a tunable manner, say in 70:30 ratio. But, this design principle was not achieved. The device can only be called programmable if within the intronic framework it shifts the balance between the two isoforms following different cellular cues/conditions. In the current scenario, it is but a static device.

2. The authors demonstrated that by engineering the intron framework, a specific isoform of the protein could be produced from an ASD. However, a major limitation is that to achieve exclusive expression of distinct isoforms, new ASD constructs must be made. This is partly because the

design is only based on cis-sequence elements. Again, this implies a new mode and not tunable regulation of the existing construct.

3. As the authors themselves identify, the behavior of ASDs made from the same intron framework but with different exons varies extensively—likely because of the introduction of cryptic splice sites, ESEs or better/worse exon definition. This limits the broader application of the engineered intron framework. Also, the exon ends that do not concur to the consensus sequences of α -tropomyosin minigene had to be mutated, which in some instances disrupted the protein function. This along with the possible interference of ESEs/cryptic splice sites would mean extensive design necessary to obtain usable future ASDs. The authors should include optimizations/adjustments to reduce the extent of aberrant products formed due to cryptic splicing. They should demonstrate that aberrant isoforms can be removed if cryptic sites within exons are mutated. This will also specify that engineered mutations in introns are not responsible for the aberrant splicing observed.

4. The entire study shows the behavior of ASDs in HEK-293T cells. It won't be unreasonable to think that the same ASDs might behave differently in different cell types that vary in expression of specific trans-acting splicing factors. To what extent have the authors explored this possibility? Do these splice ratios hold for other cell lines as well? Or do they need to be optimized depending on the cell line being used?

5. In Figure 2, the RNA-seq data for ASD clover shows three isoforms including one with a cryptic splice site. Because all the isoforms contain uninterrupted exons 1 and 3, which collectively code for Clover, why is there a 25% decrease in normalized Clover fluorescence? Do these aberrant splice isoforms not produce a functional protein because they undergo NMD? Or is the unspliced construct transcribed less efficiently than the spliced construct?

6. The application presented in Figure 5 with the TALE-TF devices is not very convincing. The fold change for the expected binding sequences from the cognate reporter was unsatisfying. In certain cases, the fold-activation was specific, while in other cases it seemed to be non-specific.

Minor Comments:

1. It would be helpful to include some representative microscopic images giving independent indications of the mCherry/clover fluorescence ratios for the MEAS constructs as well as from the co-transfected reporters of their transcription factor MEAS experiments.

2. In Figure 6, the authors tested if modulating the intronic elements as shown before can preferentially induce different isoforms of the transcription factor which can activate expression of the BFP or mCherry reporters. While the authors did gate for viability and mCherry/BFP positive populations in the flow cytometry experiments, this does not ensure that in each positive cell all three plasmids were originally transfected. It is likely that some of the data is derived from cells which only carried one of the reporters.

3. The figure legends of figure 2 and 3 do not specify any statistical information such as p values.

Response to Reviewer Comments

We would like to thank the reviewers for their constructive comments and thoughtful suggestions for improving our manuscript. The comments were very useful in helping us revise the manuscript. Below we address each comment made by the reviewers.

Reviewer #1 (Remarks to the Author):

Overall this is an excellent and timely paper. No new experiments are recommended. I recommend acceptance after some changes in the writing to make the manuscript easier to understand and the conclusions more in evidence.

The paper fills an important gap in mammalian synthetic biology: systematic engineering of alternative splicing for the expression of protein variants and artificial genetic circuits. The paper also describes, possibly for the first time, a toolkit consisting of a graded series of splice acceptors of different strengths, such that the ratio of two spliced products can be chosen by the designer. The paper also describes methods for making such a graded series that involve making only a few constructs instead of a large library. This is quite useful and deserves a mention in the abstract.

We thank the reviewer for this suggestion and have revised the abstract accordingly.

To get their MEAS system to work, the authors appear to rely on two essential tricks: (1) placement of the alternatively spliced elements so close to each other that the intron between them is too small to allow splicing – thus enforcing mutual exclusivity; and (2) varying the strengths of the splice acceptors upstream of the alternatively spliced exons, resulting in an intermediate level of use and thus in co-expression of both protein variants. This should be clearly stated in the Abstract and elsewhere (perhaps at the beginning of the Results and/or Discussion). As it is, the term “framework” is used but there is no concise description of what the framework is.

We thank the reviewer for this suggestion and have revised the text in the abstract, introduction, and results section of the manuscript accordingly.

The manuscript is generally well written.

Major Recommendations:

1. It is hard to look at Figure 2a and understand how expression of two different fluorescent proteins can result. Where are the start codons and stop codons for each ORF? The figure is particularly confusing because the ‘Clover’ and ‘mCherry’ mRNA isoforms have a significant amount of the other color. (Apologies if I missed this in the Supplementary Information, but the main text and figures should be self-sufficient for things like this.)

There is only one ORF in the ASD devices built using our intron framework, with the start codon at the beginning of exon 1 and the stop codon at the end of exon 4. We have modified FIG. 1B and 2A, to incorporate a label to the start codon (“ATG”) within exon 1 and a label to the stop codon (“TAA”) within exon 4 of the intron framework diagram to further clarify this point.

While isoforms 1-2-4 and 1-3-4 produced by both ASD mCherry (FIG. 2B) and ASD Clover (FIG. 2C) include a truncated protein coding sequence of the other fluorescent protein, this truncated sequence does not result in activity of the other fluorescent protein as demonstrated in the flow cytometry results (FIG. 2F and FIG. 2G) and the new microscopy images (FIG. 2H and FIG. 2I). The data show that only the protein whose full protein coding sequence is included in the resulting spliced RNA transcript is functionally expressed. We have modified the manuscript text in the results section to clarify this point.

2. The manuscript describes a significant amount of data from sequencing of mRNA pools. The description of these data generally seems to assume that the relative abundance of these mRNAs results from different levels of splicing. However, an alternative explanation is that nonsense-mediated decay (e.g. see Lejeune BMB Rep. 2017; 50(4): 175-185) influences the relative levels of mRNAs. The experiments with the TALENs are designed in such a way that this is not an issue, but for the fluorescent proteins this could be an issue, at least based on the way the mRNAs are diagrammed - it is important to know where the start at stop codons are for each ORF relative to the splice sites to evaluate this issue. To be clear, I do not recommend that experiments be done to measure nonsense-mediated decay directly. This would be prohibitive and would also expand the manuscript beyond a single publishable unit.

All the ASD devices described in the manuscript are designed with only one open reading frame that has one start codon at the beginning of exon 1 and one stop codon at the end of exon 4. FIG. 1B and 2A as well as text in the results section have been updated to clarify this point.

The exons are designed to contain nucleotides in multiples of three and are in frame to mitigate the appearance of a premature termination codon (PTC) in a spliced isoform that may activate nonsense-mediated mRNA decay (NMD). That said, we acknowledge that some aberrant products that use cryptic splice sites or that retain introns might result in a PTC in the transcript, be targeted for NMD, and therefore not be detectable. Thus, we focused our scope on measuring the mRNA pools that are detectable via long-read sequencing and have added text to the results section of the manuscript to clarify this aspect of our experiment.

Minor recommendations

3. In line 102, the phrase “Second, to avoid frameshift mutations” should be changed to “Second, to avoid splicing events that result in frameshifts” (A mutation is a heritable change in DNA.)

We thank the reviewer for this suggestion and have revised the manuscript accordingly.

4. Use the word “use” instead of “utilize” throughout, just to simplify sentences that are inevitably using large word. For example, in the second sentence of the abstract, replace “underutilized” with “underused”.

We thank the reviewer for this suggestion and have revised the manuscript accordingly.

5. In the Discussion (line 435-7), the authors describe a set of situations where engineered alternative splicing could be used. Voges et al. (Journal of Biological Engineering 2013, 7:20) have already used engineered alternative splicing for diversification of localization tags. The

authors could cite this reference and try to find other examples where alternative splicing has been used in synthetic biology (which may be rare or nonexistent), and possibly rephrase the sentence to describe why MEAS would be particularly valuable in certain situations.

We thank the reviewer for this suggestion. We have incorporated the suggested reference and rephrased the sentence in the manuscript as recommended.

6. The authors should include some flow cytometry plots in their figures.

We thank the reviewer for this suggestion. We have added representative flow cytometry plots to the supplementary information.

7. In line 261, the authors could use the 3-letter code for amino acids to more clearly distinguish amino acids and nucleic acid bases, especially since some 1-letter codes for amino acids may also refer to variable bases.

We thank the reviewer for this suggestion. In the TALE literature, it is conventional to refer to RVDs by their 1-letter amino acid codes (e.g., see Zhang, et al. Nat. Biotechnol. 2011; 29, 149–153 and Moore, et al. ACS Synth Biol. 2014; 3(10): 708–716). Thus, while we understand the reviewer's concern, we believe it is clearer and more consistent with TALE literature to use the 1-letter codes for amino acids when discussing our TALE work.

In addressing the reviewer comments, we noticed that the manuscript refers to amino acids using both 3-letter and 1-letter codes. Therefore, to avoid any confusion, we revised the manuscript and supplementary figures to consistently use the 1-letter codes for amino acids. We have added text to the appropriate figure legends to clarify this choice.

Reviewer #2 (Remarks to the Author):

In this manuscript, Mathur et al. engineered an intron framework, which they claim can be used in creating programmable devices to preferentially select variable isoforms from mutually exclusive splicing events. The intron framework is constructed based on a 4-exon-3-intron rat α -tropomyosin minigene system in which exons 2 and 3 are mutually exclusive by the steric restraint in spliceosome assembly between them. By splitting the coding regions of fluorescent proteins—mCherry and Clover—as exons, they demonstrate that by altering intronic elements, they can change the amount of each mutually exclusive exon that is incorporated in the final transcripts and eventually in the coded proteins. The authors show that in the intron framework used, 1-3-4 is the predominant isoform and this is due to a well-defined branch point (BP) and polypyrimidine tract (PPT) sequence in intron 2 downstream of exon 2 leading to skipping of exon 2 and inclusion of exon 3. They also show that moving this BP and PPT from intron 2 to intron 1, as well mutating the BP of intron 2, produces 1-2-4 as the predominant isoform, along with dependence of this phenomena on the length of the PPT sequence. Finally, as a proof of concept, the authors generated a TALE transcription factor with mutually exclusive exons coding for variable DNA binding domains and showed that by switching cassettes, the transcription factor could induce expression of reporter genes carrying different DNA binding elements.

Overall, the study is scientifically sound and well executed, but at this stage contrary to author's claims, their MEAS system is not truly programmable. For instance, while their intron framework is tunable to support different ratios of expression of the two isoforms, these ratios are static depending on the specific framework utilized. It would be much more exciting if the ratios were dynamically controlled in response to intrinsic or extrinsic cues. Furthermore, designing other constructs will not be trivial as they will likely include exonic contexts which may shift the splicing ratios in unexpected ways. It is difficult to envision how these devices may be used in practical applications.

While we agree with the reviewer that dynamic control is an exciting element of programmability, it is not the only aspect of programmability. Our work demonstrates programmability in terms of recoding of exonic sequences and programming different static ratios of expression of two mutually exclusive isoforms. We believe this work provides an important foundational basis for engineering mutually exclusive alternative splicing and lays the groundwork for achieving additional layers of programmability and control in future work. We have added text to the manuscript to clarify our claims.

Major Comments:

1. The authors claim that their MEAS/ASD system allows an easy way to swap out mutually exclusive regions or domains of a protein to alter function. However, what is the advantage of using this framework as opposed to simply creating separate constructs containing the sequences of interest and bypassing splicing altogether? Doing this would completely reduce aberrantly spliced or unspliced isoforms. Currently, it is unclear as to why this platform would be advantageous over the straight forward old-school method of expressing clones of individual isoforms. Although it would make sense to design such a device if the two mutually exclusive isoforms were to be expressed simultaneously in a cell in a tunable manner, say in 70:30 ratio. But, this design principle was not achieved. The device can only be called programmable if within the intronic framework it shifts the balance between the two isoforms following different cellular cues/conditions. In the current scenario, it is but a static device.

While we agree that bypassing splicing would likely be effective for many existing applications given the current state of the art in synthetic biology, we expect that use of ASDs will ultimately increase the coding capacity of genetic devices by efficiently encoding multiple functional proteins into a single genetic construct rather than requiring the use of separate constructs to encode each. This advantage is akin to data compression in the computing space, and is heavily used in natural systems, as estimates indicate that the expression of nearly 95% of human multi-exon genes involves alternative splicing (e.g., Chen, et al. Nat Rev Mol Cell Biol. 2009. 10: 741-54). For example, by expressing two TALE-TFs from an ASD built using this MEAS intron framework, we compressed the protein coding sequence by approximately 38% from 16 kilobases to 10 kilobases. We have revised the manuscript text to more clearly describe this advantage of ASDs.

As the mechanism and regulation of alternative splicing is very complex, many modes of alternative splicing are underexplored and thus currently underused in bioengineered systems. In this work, we demonstrate an early toolkit consisting of a graded series of branchpoint (BP) and polypyrimidine tract (PPT) sequences of different strengths in a programmable MEAS intron framework, such that the ratio of two spliced products can be chosen by the designer. The device

may be tuned in a static fashion to achieve several ratios of the mutually exclusive isoforms as shown in the manuscript.

To our knowledge, this is the first demonstration of a programmable MEAS system for increasing the coding capacity of genetic devices in bioengineered systems. While aberrantly spliced isoforms are currently a byproduct of ASDs, dominant behavior emerges from correctly spliced isoforms in a majority of the devices demonstrating the ASDs are capable of producing the intended isoform. We believe these results lay the groundwork for more nuanced programmability and sophisticated control in the future.

2. The authors demonstrated that by engineering the intron framework, a specific isoform of the protein could be produced from an ASD. However, a major limitation is that to achieve exclusive expression of distinct isoforms, new ASD constructs must be made. This is partly because the design is only based on cis-sequence elements. Again, this implies a new mode and not tunable regulation of the existing construct.

We appreciate this feedback. While the devices are not dynamically tunable *in vivo*, the ability to simultaneously produce two mutually exclusive isoforms via MEAS at a specific ratio is useful and is a novel contribution to the field. The intron framework is programmable, and each ASD can be customized with tuned branchpoint (BP) and polypyrimidine tract (PPT) sequences of specific strengths before use to effectively output desired ratios of mutually exclusive isoforms.

3. As the authors themselves identify, the behavior of ASDs made from the same intron framework but with different exons varies extensively—likely because of the introduction of cryptic splice sites, ESEs or better/worse exon definition. This limits the broader application of the engineered intron framework. Also, the exon ends that do not concur to the consensus sequences of α -tropomyosin minigene had to be mutated, which in some instances disrupted the protein function. This along with the possible interference of ESEs/cryptic splice sites would mean extensive design necessary to obtain usable future ASDs. The authors should include optimizations/adjustments to reduce the extent of aberrant products formed due to cryptic splicing. They should demonstrate that aberrant isoforms can be removed if cryptic sites within exons are mutated. This will also specify that engineered mutations in introns are not responsible for the aberrant splicing observed.

We appreciate the reviewer's comment and agree that alternative splicing is a complex process with a number of confounding factors potentially influencing splice site selection including cryptic splice sites, ESEs, and varied exon definition. As a result, alternative splicing has been underused in bioengineered systems. As a starting point, our work attempts to limit the introduction of further complexity to create a framework that can be used with the application of three design criteria for programming exonic sequences, specifically (1) mutually exclusive exon lengths of approximately 126 nucleotides, (2) exon lengths in multiples of three nucleotides, and (3) splice site sequences fixed to those from the minigene to support accurate exon definition.

We recognize that ESEs do affect splicing outcomes because ASD mCherry, ASD Clover, and each ASD TALE-TF contain unique cryptic splice sites that produce different aberrant isoforms. All of the engineered intronic mutations were tested in the ASD mCherry devices and their RNA isoforms did not reveal cryptic splice sites in exons 1 or 4, while cryptic splice sites did arise in other devices (e.g., ASD Clover and ASD TALE-TF). This outcome suggests that the

engineered intronic mutations alone are not responsible for the observed aberrant splicing. According to our long-read sequencing experiments, aberrant isoforms that arise from cryptic splice sites in the major devices (e.g., ASD mCherry, ASD Clover, ASD TALE 1 T/G, and ASD TALE 2 TT/AA) account for only ~8.5% of the output. Across all devices, which include ones that harbor engineered intronic mutations, the relative abundance of aberrant products that result from cryptic splice sites varies and at most accounts for approximately a quarter of total spliced transcripts.

Nonetheless, we conducted preliminary optimizations in ASD Clover attempting to remove two cryptic splice sites in exon 4 that were identified early in our experiments with some success. Specifically, in this preliminary effort we were able to remove one of the cryptic splice sites, but not the other cryptic splice site because efforts to remove it resulted in the emergence of a novel cryptic splice site in exon 4 (data not shown). Due to the number of known and unknown confounding factors and their interactions, optimizations geared toward effective cryptic splice site removal were not immediately illuminating as they could result in unanticipated downstream effects, such as the emergence of novel cryptic splice sites. Therefore, we focused our efforts on systematically characterizing a diverse set of ASDs built using the MEAS intron framework to comprehensively catalogue cryptic splice sites and aberrant products in a manner that may support future efforts to mitigate undesired outputs. Our work demonstrates that despite the variable exon sequences across ASDs and the presence of cryptic splice sites, a large fraction of the transcripts generated by these devices splice to the intended mutually exclusive isoforms. We have added text to the discussion section of the manuscript to include a note about our early optimization efforts.

4. The entire study shows the behavior of ASDs in HEK-293T cells. It won't be unreasonable to think that the same ASDs might behave differently in different cell types that vary in expression of specific trans-acting splicing factors. To what extent have the authors explored this possibility? Do these splice ratios hold for other cell lines as well? Or do they need to be optimized depending on the cell line being used?

We thank the reviewer for this question. We focused on thoroughly characterizing the activities of these ASDs in one cell type (HEK-293T) because prior studies have shown that the α -tropomyosin minigene splices by default to isoform 1-3-4 in all cell types other than smooth muscle cells (e.g., Gooding, et al. EMBO J. 1994. 13:3861–72). Thus, while we expect that other cell types may result in some variation, the selected HEK-293T cell type should be representative of a significant cross section of cell types.

To address the reviewer's question, we have performed new experiments that characterize a subset of the reported ASD mCherry devices in three new cell lines (HeLa, CHO-K1, and U2OS). The new experimental results demonstrate that the splicing outcomes and trends produced by the ASD mCherry devices in the new cell lines are comparable and within error to those observed in HEK-293T. These results suggest that ASDs could be used reliably in different cell types without significant optimization. The minor differences in fluorescence levels that are observed across cell lines probably result from differences in the composition and abundance of splicing regulators in these cells. These new results have been added to the supplementary information and some text describing this experiment has been added to the results section of the manuscript.

5. In Figure 2, the RNA-seq data for ASD clover shows three isoforms including one with a cryptic splice site. Because all the isoforms contain uninterrupted exons 1 and 3, which

collectively code for Clover, why is there a 25% decrease in normalized Clover fluorescence? Do these aberrant splice isoforms not produce a functional protein because they undergo NMD? Or is the unspliced construct transcribed less efficiently than the spliced construct?

We thank the reviewer for this question. We are not completely certain as to why Clover expression is diminished in the ASD Clover devices. Since there is only one open reading frame in this device, we expect most spliced isoforms do not undergo NMD. We speculate that these aberrant products may be transcribed less efficiently than the spliced control or that the mCherry coding sequence in exon 4 may produce a peptide that interferes with proper folding and maturation of Clover. We have added text clarifying this point to the results section of the manuscript.

6. The application presented in Figure 5 with the TALE-TF devices is not very convincing. The fold change for the expected binding sequences from the cognate reporter was unsatisfying. In certain cases, the fold-activation was specific, while in other cases it seemed to be non-specific.

We thank the reviewer for this comment. The TALE-TFs used in this work were selected from a highly cited paper in the TALE field (Zhang, et al. Nat. Biotechnol. 2011. 29: 149–53) that measured the relative activity of full-length TALE-TF sequences in activating fluorescent reporters from binding sites harboring one or two nucleotide mismatches. We chose TALE-TFs that had been tested with binding sites with mismatches in the middle of their sequences so that we could incorporate the corresponding RVDs into the internal mutually exclusive exons. The fold activations presented in this work are within previously established values in the literature for fold activations from transient transfection reporter assays (e.g., Meckler, et al. Nuc Acids Res. 41:4118-28). There is also evidence from previous studies that supports that disrupting binding at the termini of the binding sequence has less of an effect on activation than disrupting binding in the middle of the sequence (e.g., Rogers, et al. Nat. Comm. 2015. 6:7440) and that some TALE-TFs prefer to bind a mismatched sequence rather than the sequence the RVD code predicts. For example, the NN-targeting RVD can bind both nucleotides G and A (e.g., Boch, et al. Science. 2009. 326:1509-12), with other RVDs only slightly preferring a non-target nucleotide.

In our study, we conducted a systematic assessment of several combinations of TALE-TFs and binding site sequences. Given a single nucleotide mismatch at the termini of the binding sequence, we would expect some off-target activation from the substituted RVD. We observe more specificity with a dinucleotide mismatch in the middle of the binding sequence. Overall, our analysis with long-read sequencing and fold activation assays was sufficient to demonstrate that synthetic transcription factors can be built within the ASD framework and produce fluorescent reporter activation with fold activations comparable to their full-length controls.

Minor Comments:

1. It would be helpful to include some representative microscopic images giving independent indications of the mCherry/clover fluorescence ratios for the MEAS constructs as well as from the co-transfected reporters of their transcription factor MEAS experiments.

We thank the reviewer for this suggestion. To address this comment, we have added fluorescence microscopy images of ASD mCherry and ASD Clover to FIG. 2. We have also added representative fluorescence microscopy images of ASD mCherry and ASD TALE-TF 1 T/G

devices harboring modified BP and PPT elements to the supplementary information and some text describing these images to the results section of the manuscript.

2. In Figure 6, the authors tested if modulating the intronic elements as shown before can preferentially induce different isoforms of the transcription factor which can activate expression of the BFP or mCherry reporters. While the authors did gate for viability and mCherry/BFP positive populations in the flow cytometry experiments, this does not ensure that in each positive cell all three plasmids were originally transfected. It is likely that some of the data is derived from cells which only carried one of the reporters.

We thank the reviewer for this comment. In analyzing the data and calculating fold activations, we gated for one reporter plasmid at a time and verified that given the presence of that reporter plasmid, cells were usually also expressing the other reporter plasmid. We only gate for one color at a time because we cannot assume that both mCherry and Clover would be activated in a cell by the co-transfected ASD TALE-TF or TALE-TF control. For example, in measuring fold activations from the BFP reporter plasmid, we gate for BFP-positive cells. When we assess the percentage of BFP-positive cells that express mCherry, we routinely see ~75-95% overlap. In the reverse scenario, we also observe ~75-95% of mCherry-positive cells express BFP. The newly-added fluorescence microscopy images and flow cytometry plots for ASD TALE-TF 1 T/G in the supplementary information also illustrate that we routinely observe both reporter plasmids in the transfected cell populations. We agree that in our population-level analysis, we may include a small fraction of cells that contain only one reporter or two reporters at differing copy numbers, but we use median expression levels of the transfected population to reflect the overall fold activations from the system as a resistant statistic of central tendency.

Clover is a transfection marker for the ASD TALE-TFs and TALE-TF controls. Given the relative amounts of the ASD TALE-TFs or TALE-TF controls (700 ng) versus the reporter plasmids (50 ng per reporter) in the transient transfections, we prioritized ensuring that the mCherry/BFP-positive cells were gated for appropriately in our fold activation calculations. We had initially planned to use the 2A-Clover fluorescent tag on the ASD-TALE-TF devices to gate for Clover-positive cells. However, with these constructs, we observed dim expression of Clover that made it difficult to compare between TALE-TF controls and the ASD TALE-TF devices. To avoid biasing the results, we only gated for mCherry/BFP-positive cells as a conservative measure of fold activation; as a result, we expect that we are underestimating fold activation in our calculations.

3. The figure legends of figure 2 and 3 do not specify any statistical information such as p values.

The manuscript presents statistical information in the form of standard error. Neither FIG. 2 nor FIG. 3, nor their descriptions in the manuscript, attempt to support or test significance of any particular statistical hypothesis, so p values are not specified.

Reviewers' Comments:

Reviewer #2:

Remarks to the Author:

Thank you for responding to my comments by carrying out additional experiments and/or providing stronger rationale and arguments throughout the paper. The revised manuscript is much stronger and provides a nice tool to study mutually exclusive splicing in living cells.